# nNOS-expressing interneurons control basal and behaviorally evoked arterial dilation in somatosensory cortex of mice

Christina T Echagarruga[1], Kyle W Gheres[2], Jordan N Norwood[3], Patrick J Drew[1,2,3,4]*

[1]Bioengineering Graduate Program, Pennsylvania State University, University Park, United States; [2]Molecular, Cellular, and Integrative Biology Graduate Program, Pennsylvania State University, University Park, United States; [3]Cell and Developmental Biology Graduate Program, Pennsylvania State University, University Park, United States; [4]Departments of Engineering Science and Mechanics, Biomedical Engineering, and Neurosurgery, Pennsylvania State University, University Park, United States

**Abstract** Cortical neural activity is coupled to local arterial diameter and blood flow. However, which neurons control the dynamics of cerebral arteries is not well understood. We dissected the cellular mechanisms controlling the basal diameter and evoked dilation in cortical arteries in awake, head-fixed mice. Locomotion drove robust arterial dilation, increases in gamma band power in the local field potential (LFP), and increases calcium signals in pyramidal and neuronal nitric oxide synthase (nNOS)-expressing neurons. Chemogenetic or pharmocological modulation of overall neural activity up or down caused corresponding increases or decreases in basal arterial diameter. Modulation of pyramidal neuron activity alone had little effect on basal or evoked arterial dilation, despite pronounced changes in the LFP. Modulation of the activity of nNOS-expressing neurons drove changes in the basal and evoked arterial diameter without corresponding changes in population neural activity.

**\*For correspondence:**
PJD17@PSU.EDU

**Competing interests:** The authors declare that no competing interests exist.

## Introduction

Understanding the mechanisms by which neural activity regulates local blood flow is a fundamental issue in neuroscience. Changes in neural activity are coupled to hemodynamics via neurovascular coupling, where neural activity drives the dilation of arterioles and other vessels (*Drew et al., 2011*; *Hillman, 2014*; *Hill et al., 2015*; *Mishra et al., 2016*; *Iadecola, 2017*; *Rungta et al., 2018*; *Drew, 2019*). Arterial dilations are most strongly correlated to increases in the gamma-band power of the LFP (local field potential), with lower correlation to the power in other bands of the LFP (*Goense and Logothetis, 2008*; *Logothetis, 2008*; *Mateo et al., 2017*; *Winder et al., 2017*; *Drew et al., 2020*). While it is controversial whether capillaries dilate (*Hall et al., 2014*; *Hill et al., 2015*; *Mishra et al., 2016*; *Rungta et al., 2018*) and whether this dilation is active or passive (*Mishra et al., 2016*; *Rungta et al., 2018*), sensory stimulation drives reliable and large (~20%) dilations of cerebral arteries in un-anesthetized animals (*Drew et al., 2011*; *Hill et al., 2015*; *Rungta et al., 2018*; *Drew, 2019*), which contributes to increases in blood flow. The sensory-evoked dilation initiates in the deeper layers of the cortex (*Tian et al., 2010*; *Rungta et al., 2018*), and rapidly propagates up to the surface (pial) arteries (*Drew et al., 2011*; *Mateo et al., 2017*) via conduction of an electrical signal (*Chen et al., 2011*; *Hillman, 2014*; *Longden et al., 2017*). Pial arteries show dilations correlated with spontaneous gamma band power (*Mateo et al., 2017*), and in awake animals, pial arteries show stimulus-selective dilations in response to sensory stimulation

(*Drew et al., 2011*; *Gao and Drew, 2016*; *Winder et al., 2017*). Pial arteries in the histologically reconstructed forelimb/hindlimb representation of primary somatosensory cortex dilate substantially more during locomotion than arteries just outside of this representation (*Gao et al., 2015*), although there may be less sensory specificity of the pial vessels responses in anesthetized animals (*O'Herron et al., 2016*). Simulations have suggested that the dilation of these arteries can play an important role increasing blood flow during functional hyperemia (*Schmid et al., 2017*; *Rungta et al., 2018*).

There are multiple signaling pathways and molecules implicated in linking neural activity (both directly and indirectly) to arteriole dilations (*Attwell et al., 2010*; *Cauli and Hamel, 2010*; *Kleinfeld et al., 2011*; *Lacroix et al., 2015*; *Longden et al., 2017*). While no single mechanism likely underlies neurovascular coupling, there is an extensive literature connecting the vasodilator nitric oxide (NO) to vasodilation both in the periphery and the brain (*Tanaka et al., 1991*; *Iadecola, 1992*; *Adachi et al., 1994*; *Stefanovic et al., 2007*; *Hosford and Gourine, 2019*; *Han et al., 2019a*). Recent in vitro evidence indicates neuronally-generated nitric oxide acts directly on arteries, while astrocytic signals control capillary dilations (*Mishra et al., 2016*). However, which set(s) of neurons control arterial diameter is not well understood. Optogenetic stimulation of interneurons drives a larger vasodilation than optogenetic stimulation of pyramidal neurons (*Vazquez et al., 2014*; *Anenberg et al., 2015*; *Uhlirova et al., 2016b*; *Vazquez et al., 2018*; *Krawchuk et al., 2020*), suggesting a subset of neurons exerts a disproportionate control over vasodilation. Optogenetic stimulation of neuronal nitric oxide (nNOS, also known as NOS1) expressing interneuron, but not in VIP or parvalbumin interneurons, produces increase in blood flow and vasodilation (*Krawchuk et al., 2020*; *Lee et al., 2020*), and this vasodilation is not accompanied by a detectable increase in neural activity (*Lee et al., 2020*), suggesting that nNOS neurons can influence local hemodynamics without detectably altering the activity of the rest of the network.

There are two types of nNOS–expressing neurons in the cortex, Type 1 and Type 2 (*Kawaguchi and Kubota, 1997*; *Vruwink et al., 2001*; *Kubota et al., 2011*). Type 1 nNOS neurons are sparse, are primarily found in deeper layers and express nNOS at higher levels. While Type 1 nNOS neurons receive many different neuromodulatory inputs (*Williams et al., 2019*; *Williams et al., 2018*), nearly all of them express the Substance P (SP) receptor (NK1R), and both single-cell genomic and histological studies have shown that they are the only cells in the mouse cortex that express the Substance P receptor (*Vruwink et al., 2001*; *Dittrich et al., 2012*; *Barbaresi et al., 2015*; *Endo et al., 2016*; *Tasic et al., 2016*). In vitro studies have shown that application of Substance P causes prolonged high-frequency spiking in type 1 nNOS neurons (*Dittrich et al., 2012*; *Endo et al., 2016*). Type 1 nNOS neurons also have extensive, long-range projecting axonal arbors that well-position them to influence the vasculature. In contrast, type 2 nNOS neurons are a heterogeneous group of interneurons that are substantially more numerous, found primarily in the deep and superficial layers of the cortex, with processes that can span multiple layers (*Perrenoud et al., 2012*). Type 2 nNOS neuron express nNOS at lower levels than type 1 nNOS neurons. The synthesis of NO by nNOS is calcium dependent, and as NOS is found both in the cell bodies and the processes of neurons NO will be produced throughout the cell when calcium levels rise with activity (*Valtschanoff et al., 1993*; *Blottner et al., 1995*).

While most studies of neurovascular coupling have focused on the impact of brief elevations of neural activity on hemodynamic signals, the role of neural activity in setting the baseline levels of these signals is not well understood, but have important implications for understanding the relationship between neural activity and hemodynamic signals. Most hemodynamic imaging modalities, such as BOLD fMRI, laser speckle, laser Doppler and Doppler ultrasound, are ratiometric, meaning they measure a percentage change from a baseline rather than an absolute change. This means that these measurements are sensitive to the baseline level of flow and dilation. For a given absolute change in flow, an elevated baseline will reduce the amplitude of the signal. This issue has long been appreciated in fMRI (*Cohen et al., 2002*; *Whittaker et al., 2016*; *Kim et al., 2020*), and there are techniques available to calibrate these signals (*Kim and Ogawa, 2012*). Differences in baseline are an important confound in comparisons across different groups, confounding aging and disease studies in humans (*D'Esposito et al., 2003*), particularly as in humans baseline blood flow drops with aging (*Ruitenberg et al., 2005*; *Wolters et al., 2017*). However, the impact of shifting the baseline levels, and particularly how neural activity could affect them, has been underappreciated in neurovascular coupling studies, which often compare the impact of disrupting a pathway in different

groups of animals using ratiometric techniques. This is particularly a problem because it is not well understood how baseline (non-stimulus evoked) levels of neural activity control blood flow, and ongoing neural activity in the un-anesthetized brain is very high. In contrast, two-photon imaging can measure absolute hemodynamic signals (*Shih et al., 2012b*), such as dilations of single vessels and velocities (*Kleinfeld et al., 1998*) and fluxes (*Chaigneau et al., 2003*; *Shih et al., 2009a*) of red blood cells.

Elucidating which neuronal cell types control arterial dilation (*Kleinfeld et al., 2011*; *Uhlirova et al., 2016a*) is important for interpreting hemodynamic imaging signals (*Logothetis, 2008*; *Kim and Ogawa, 2012*). Moreover, the regulation of *basal* arterial diameter, which plays a role in the control of baseline blood flow (*Gagnon et al., 2015*; *Schmid et al., 2017*), is also important because decreases in basal cerebral blood flow occur with aging, and the magnitude of these decreases predicts neurodegeneration (*Ruitenberg et al., 2005*; *Wolters et al., 2017*).

## Results

We used two-photon microscopy (*Shih et al., 2012b*) to chronically image pial and penetrating arterioles in the somatosensory cortex of awake, head-fixed mice (*Figure 1A and B*; *Drew et al., 2011*; *Gao and Drew, 2016*; *Winder et al., 2017*) through polished and reinforced thinned-skull (PoRTs) windows (*Drew et al., 2010b*), which allow chronic imaging without inflammation. Because we were able to repeatedly image from the same vessels over multiple imaging sessions (*Figure 1C*), we were able to make quantitative comparisons of individual vessel diameter changes in response to manipulations of neural activity, rather than compare vessel responses from two different groups of animals. All drug/CNO, vehicle infusions, and injections were randomly counterbalanced across animals. As we measured from multiple vessels in each mouse, we used linear mixed-effect models (LME, see Materials and methods) (*Aarts et al., 2014*) for statistical tests to account for within-animal correlations (*Winder et al., 2017*; *Adams et al., 2018*) that make methods such as ANOVA inappropriate (*Aarts et al., 2014*).

We measured arteriole diameter dynamics evoked by locomotion, as behavior is the primary driver of neural and hemodynamic signals in the awake brain (*Figure 1D*; *Gao and Drew, 2016*; *Winder et al., 2017*; *Stringer et al., 2019*), and because sensory stimulation in awake animals invariably elicits movement (*Cooke et al., 2015*; *Drew et al., 2019*; *Musall et al., 2019*; *Salkoff et al., 2020*). The change in diameter and neural activity during locomotion was quantified by averaging the dilation or neural activity 3 to 4 s after the onset of locomotion. We reconstructed the locations of individual arterioles relative to the forelimb/hindlimb (FH/HL) representation in somatosensory cortex (purple and green shaded regions *Figure 1B*, top), as arteries inside the FL/HL region dilate selectively in response to locomotion, significantly more so than arteries immediately outside the histologically-identified FL/HL (*Gao et al., 2015*). Locomotion-induced arterial dilations in the FL/HL area of somatosensory cortex have highly reliable and stereotyped dynamics (*Huo et al., 2015b*), and are unaffected by pharmacological manipulations of heart rate and blood pressure (*Huo et al., 2015a*), consistent with local neural control. During bouts of voluntary locomotion, neural activity increased substantially in the forelimb/hindlimb representation in somatosensory cortex, as measured by changes in multi-unit firing rate and power in the gamma-band of the LFP (*Figure 1—figure supplement 1*; *Huo et al., 2014*; *Zhang et al., 2019*). These increases are driven primarily by cutaneous sensation rather than proprioception (*Chapin et al., 1981*; *Chapin and Woodward, 1981*; *Chapin and Woodward, 1982*; *Chapin and Lin, 1984*). During locomotion, this neural activity is sustained, and shows minimal adaptation (*Zhang et al., 2019*). In response to this increase in neural activity, both pial and penetrating arterioles in the FL/HL representation dilated (*Huo et al., 2015b*; *Gao and Drew, 2016*; *Figure 1D* and *Figure 1—figure supplement 2* and *Video 1* and *2*) with short latency (onset time: 0.59 ± 0.31 s in the FL/HL region).

### Basal arterial diameter and evoked dilation are controlled by local neural activity

We first determined the role of overall local neural activity in maintaining baseline arterial diameter. We refer to the diameter of the vessel during periods lacking locomotion (see Materials and methods) as the 'basal' diameter. We suppressed the activity of all neurons via intracranial infusions of muscimol (a GABA$_A$ receptor agonist) and compared the changes in neural activity and arterial

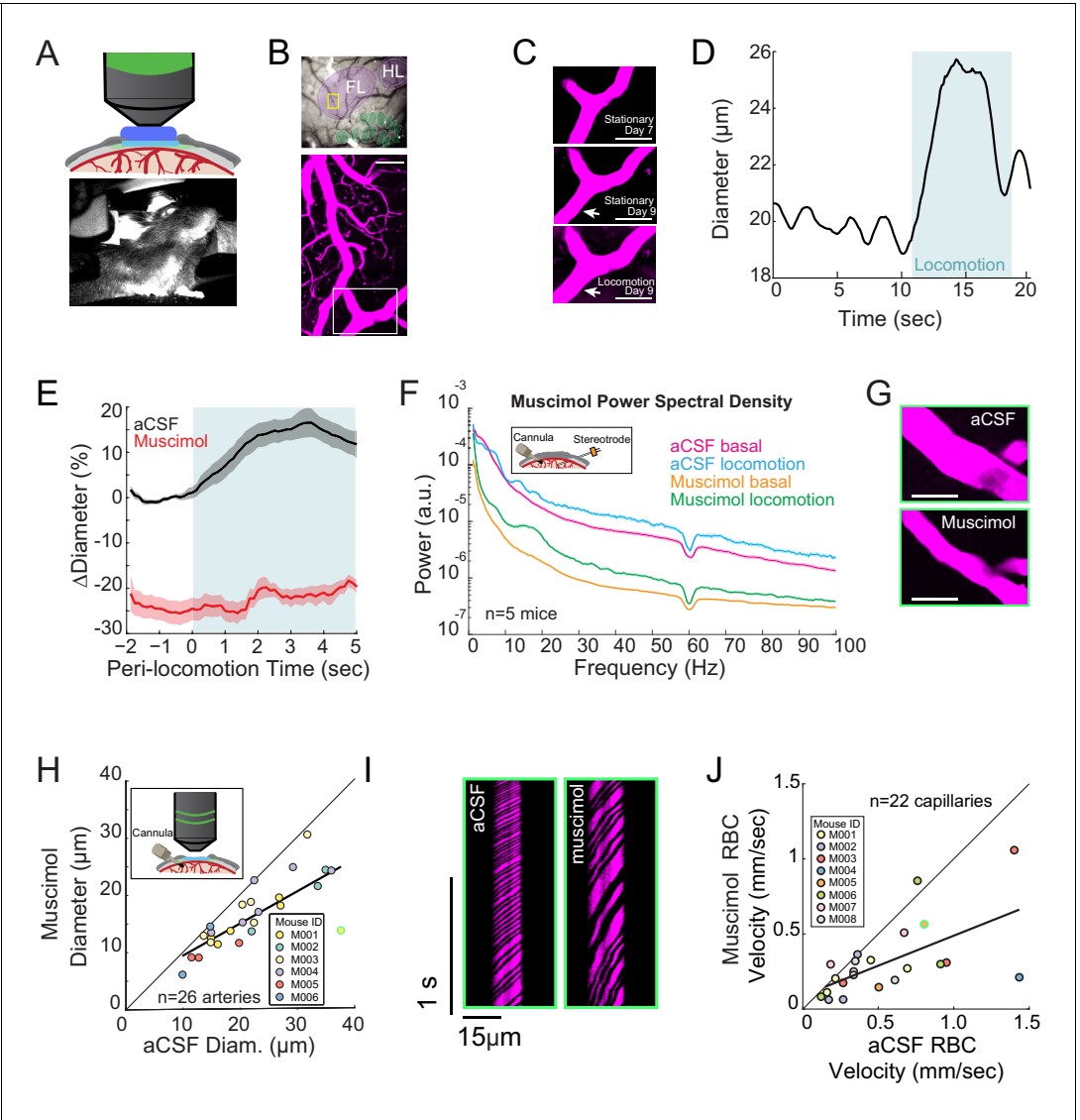

**Figure 1.** Local neural activity controls basal and evoked arteriole diameter. (**A**) Top, schematic of imaging window. Bottom, photo of mouse on spherical treadmill. (**B**) Top, photo of the pial vasculature of the somatosensory cortex through the PoRTs window. Cytochrome oxidase staining localized the forelimb/hindlimb (FL/HL, purple) and vibrissae cortex (green). Bottom, a maximum projection of two-photon images of vasculature within the yellow box in the top image. Scale bar 50 μm. (**C**) Top and middle, two-photon images (from white box in B, bottom) of the same arteriole 7 and 9 days after window implantation, both from stationary periods. Bottom, arteriole 9 days after implantation during locomotion. Scale bar 50 μm. (**D**) Locomotion induces rapid dilation in pial arterioles. Diameter (from region marked by white arrow in **C**) plotted versus time for a single locomotion event. Blue shading denotes period of locomotion. (**E**) Population locomotion-triggered averages following aCSF and muscimol infusions for arterioles ≤ 25 μm in basal diameter. For both cases, the individual diameters are normalized by the average basal diameter of the vessel after vehicle infusion. Note the rapid rise to peak in the control, and the lack of dilation in muscimol-infused mice, showing the locomotion-triggered response is under local neural control. Shading represents mean ± standard error. (**F**) LFP power spectra during stationary periods (basal) and locomotion after muscimol infusion, normalized to vehicle infusion in the same animal. Shading represents mean ± standard error. (**G**) Representative images of a pial arteriole during periods of no locomotion after vehicle infusion (top) and after muscimol infusion (bottom). Scale bar 50 μm. (**H**) Basal arteriole diameter following vehicle infusion plotted versus basal diameter after muscimol infusion. Each point is a single vessel, and the mouse identity is represented by the color. The black line shows the linear regression of aCSF vs. muscimol diameter. The point outlined in green is representative image from G. (**I**) Representative space-time image of linescans of the same capillary after aCSF or muscimol infusion. (**J**) Basal red blood cell (RBC) velocity plotted after vehicle infusion (x-axis) vs. after muscimol infusion (y-axis). The point outlined in green is the vessel shown in **I**.

The online version of this article includes the following figure supplement(s) for figure 1:

**Figure supplement 1.** Example neural responses in forelimb/hindlimb representation of somatosensory cortex during voluntary locomotion.

**Figure supplement 2.** Diameter changes in penetrating arterioles.

**Figure supplement 3.** Muscimol infusion did not significantly affect capillary diameter.

*Figure 1 continued on next page*

*Figure 1 continued*

**Figure supplement 4.** Muscimol infusion did not significantly affect basal arteriole diameter variance.
**Figure supplement 5.** Effects of chemogenetic and pharmacological manipulation on locomotion behavior.

diameter dynamics to artificial cerebrospinal fluid (aCSF- a vehicle control) infusions in the same mouse. Infusions of muscimol substantially decrease (by 70–95%) neural activity ~1.5 mm from the cannula (*Winder et al., 2017*; *Zhang et al., 2019*), and so will suppress activity in all cortical layers, unlike topical administration of drugs which typically only affect the superficial layers (*Ferezou et al., 2006*). Control experiments have shown that cannula implantations do not alter hemodynamic responses (*Winder et al., 2017*). To visualize any change in vessel diameter across all the measured vessels we made a scatter plot of vessel diameters (*Figure 1G–H*), with the abscissa the diameter measured after the control (vehicle) infusion, and the ordinate the diameter after the drug infusion (muscimol in this case). If there is no effect of the drug, the points will be scattered along the unity line. If the drug causes constriction or dilation, the points will be respectively above or below the line. We found that after the infusion of muscimol, there was a substantial decrease in baseline and locomotion-evoked gamma-band power relative to within animal aCSF controls (*Figure 1F*, basal $-75.2 \pm 7.1\%$, $p<3.0\times10^{-4}$; locomotion, $-77.0 \pm 6.6\%$, $p<2.0\times10^{-4}$). We found that muscimol infusions led to a substantial decrease in the basal arteriole diameter (*Figure 1G–H*, $-25.4 \pm 2.3\%$, LME $p<2.8\times10^{-4}$, n = 6 mice, 26 arterioles; *Figure 1—figure supplement 2B*). The effects of muscimol infusions were not due to direct actions on the cerebral vasculature, as multiple large-scale single-cell sequencing studies have shown that mice lack GABA receptors on endothelial cells (*Sabbagh et al., 2018*; *Vanlandewijck et al., 2018*; *Zeisel et al., 2018*) and physiological experiments using optogenetic stimulation of interneurons with and without GABA-receptor blockade have shown there is no impact of blocking GABAergic signaling on vessel dilation (*Anenberg et al., 2015*; *Vazquez et al., 2018*). Rather than being driven by direct action of muscimol on the vasculature, this constriction is likely due to the removal of tonic vasodilatory signals released by neurons.

To determine if this decrease in basal arteriole diameter drove a decrease in blood flow, we then measured red blood cell (RBC) velocity in the capillaries (*Kleinfeld et al., 1998*; *Chaigneau et al., 2003*; *Drew et al., 2010a*). Capillary RBC velocity was markedly decreased in muscimol-infused mice relative to aCSF infusions (*Figure 1I and J*, $-34.5 \pm 27\%$, LME $p<1.5\times10^{-3}$, n = 8 mice, 22 capillaries). We observed no significant change in the diameter of the capillaries following the infusion of muscimol (aCSF-muscimol difference: $0.03 \pm 0.85$ µm, LME $p<0.94$, *Figure 1—figure supplement 3A and B*). Suppressing neural activity with muscimol did not abolish spontaneous fluctuations in arteriole diameter (*Figure 1—figure supplements 4A*, -14.2±67%, LME $p<0.06$, n = 6 mice, 26 vessels), consistent with previous reports of vessel-autonomous oscillations in arterioles (*Winder et al., 2017*). Taken together, these results suggest that baseline levels of neural activity have a tonic vasodilatory effect on cerebral arterioles (either directly, or through astrocytes), and decreasing neural activity causes a corresponding decrease in arteriole diameter and blood flow.

We then examined the role of neural activity in locomotion-evoked dilations. We calculated locomotion-triggered averages using only the arterioles histologically identified to be in the FL/HL region (using cytochrome oxidase staining [*Adams et al., 2018*]), as vessels outside this region are less responsive to locomotion (*Gao et al., 2015*). We also restricted this locomotion-triggered analysis to arterioles < 25 microns in diameter, as arterial reactivity decreases with increasing arteriole size (*Lee et al., 2001*; *Drew et al., 2011*; *Gao et al., 2015*), so larger vessels tend to be less

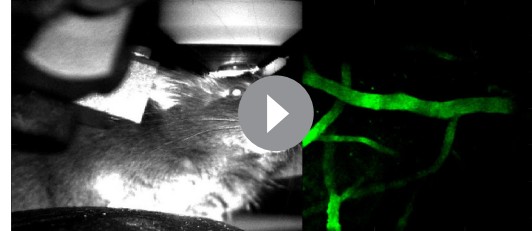

**Video 1.** Locomotion produces a rapid dilation in pial arterioles. This movie shows in vivo imaging of a surface arteriole diameter in the somatosensory cortex through the PoRTs window using two-photon microscopy. Left, behavioral camera. Right, FITC-filled arteriole diameter (region marked by white arrow) was plotted versus time in green.
https://elifesciences.org/articles/60533#video1

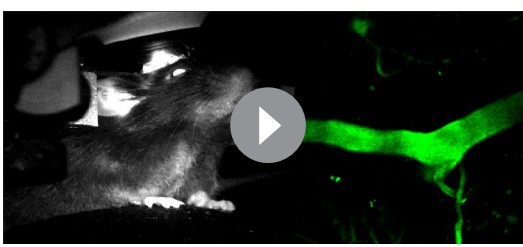

**Video 2.** Locomotion produces a rapid dilation in pial arterioles. This movie shows in vivo imaging of a surface arteriole diameter in the somatosensory cortex through the PoRTs window using two-photon microscopy. Left, behavioral camera. Right, FITC-filled arteriole diameter (region marked by white arrow) was plotted versus time in green.
https://elifesciences.org/articles/60533#video2

responsive to changes in neural activity than smaller vessels. We found that the locomotion-evoked dilation of these arterioles was blocked by local silencing of neurons with muscimol (*Figure 1E* locomotion-evoked dilation: aCSF 15.8 ± 2.7%; muscimol −21.6 ± 1.9%, LME p<5.2×10$^{-6}$, n = 6 mice, 12 arterioles in FL/HL representation), suggesting that these arteriole dilations were not due to non-specific cardiovascular responses (*Huo et al., 2015a*) or modulatory input from other brain regions directly onto the vessels (*Drew et al., 2008*), but rather controlled by local activity. Similar effects were observed in penetrating arterioles (*Figure 1—figure supplement 2B and C*; see also *Figure 6—figure supplement 2*). Note that the locomotion-evoked numbers reported in the text (unless otherwise stated), are relative to the control baseline, and thus include both the effects of the manipulation on the basal diameter and the evoked diameter. We split out the effects of our manipulations on the locomotion-evoked response in the analysis in *Figure 6—figure supplement 2*. For comparison of the results obtained with these two approaches, during muscimol infusions, there was a small locomotion-evoked increase from the pre-locomotion baseline (15.37 ± 8.11% vs. 5.14±11.09%, LME p<0.001, n = 5 animals, 12 vessels) (*Figure 6—figure supplement 2*). Infusions of muscimol and other manipulations of neural activity did not cause significant differences in the amount of locomotion relative to controls, indicating that behavior is not an origin of this difference (*Figure 1—figure supplement 5*), and consistent with previous work showing that pharmacological silencing of motor cortex disrupted complex movements, but not locomotion (*Beloozerova and Sirota, 1993*). Thus, both basal diameter and sensory-evoked arterial dilations in the FL/HL region of somatosensory cortex are under control of local neural activity.

## Bidirectional chemogenetic manipulation of local neural activity controls basal arterial diameter

We then asked if we could bi-directionally manipulate arterial diameters using chemogenetic techniques to drive changes in overall neural activity. We expressed either hM4D-G(i) or hM3D-G(q)-coupled DREADDs (*Urban and Roth, 2015*) pan-neuronally under a human synapsin promoter (hSyn) in C57BL/6J mice. G(i)- and G(q)-coupled DREADDs will respectively decrease and increase the excitability of the cells expressing them after injection of the DREADDs agonist CNO. As a control, mice infected with an AAV encoding only the fluorescent reporter mCherry did not show significant changes in neural activity, or in basal or evoked arterial diameter after the injection of CNO (*Figure 2—figure supplement 1B–D*), showing CNO has minimal off-target effects on neural activity and arterial diameters.

We imaged arterioles in the DREADD-expressing region of cortex (identified by mCherry expression seen in vivo, *Figure 2A*, and histologically, *Figure 2B*). Injections of CNO in mice with pan-neuronal expression of hM4D-G(i) DREADDs resulted in a non-significant decrease in gamma-band power relative to vehicle injections (*Figure 2C*, basal −1.9 ± 27.2%, p<0.09; locomotion, +3.6 ± 10.3%, p<0.07), likely because of the parallel changes in excitability of inhibitory and excitatory cells cancelled out. Still, activation of hM4D-G(i) DREADDs resulted in significant decreases in basal arterial diameter (*Figure 2D*, −7.1 ± 7.0%, LME p<0.02 n = 7 mice, 45 vessels), which was particularly noticeable in the largest arterioles. We note that the hM4D-G(i) DREADDs did not uniformly lower neural activity, as we observed bursting activity in our electrophysiological recordings every few seconds (*Figure 2—figure supplement 2A*), and corresponding arterial dilations at similar frequencies (*Figure 2—figure supplement 2B*). While we saw clear constrictions in arteriole diameters when activating hSyn G(i)-DREADDs relative to baseline, there were also large spontaneous dilations, which on average cancelled out. The net effect of hM4D-G(i) DREADDs on the locomotion-evoked

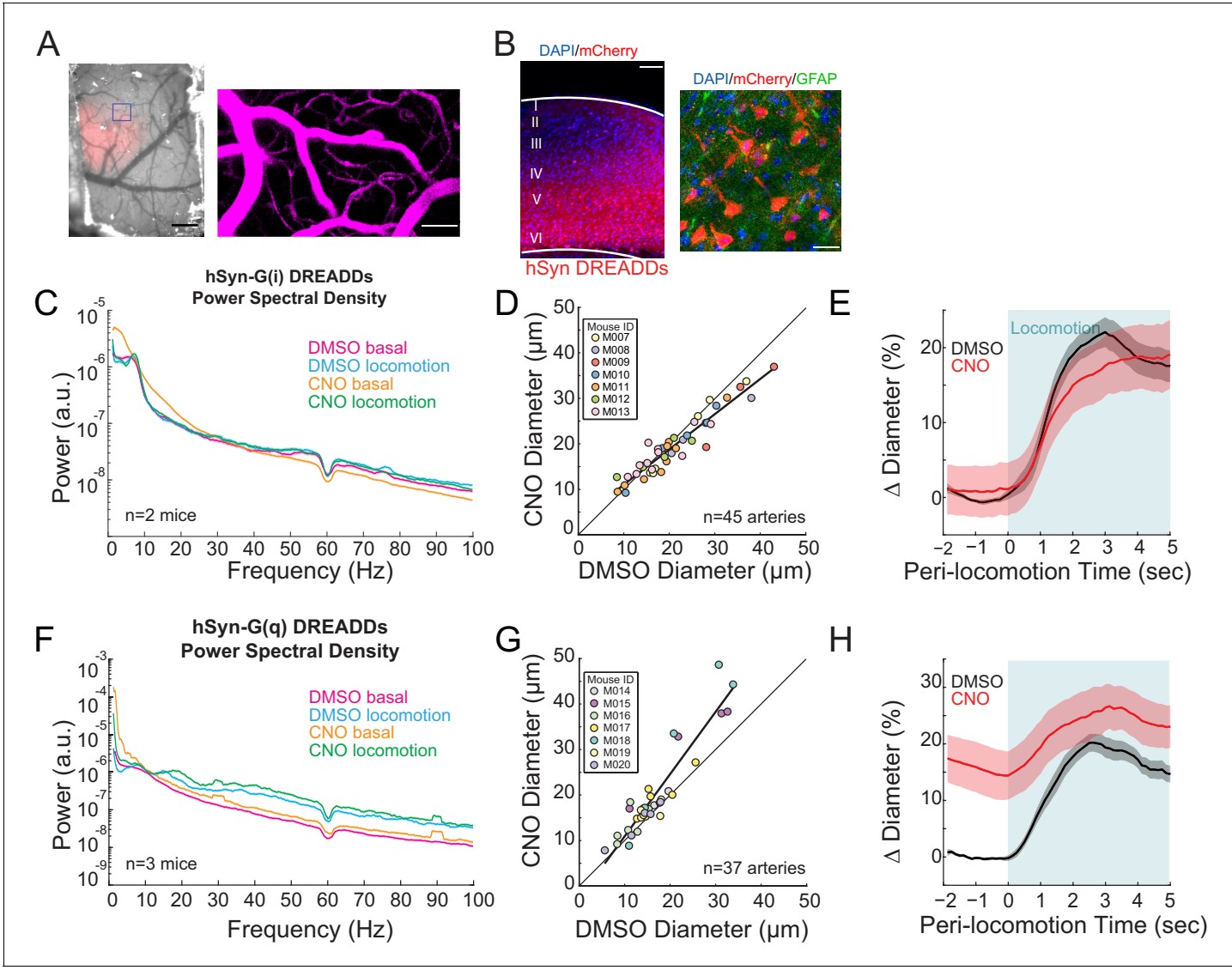

**Figure 2.** Neural activity bidirectionally controls basal arteriole diameter. (**A**) Left, image through the polished and reinforced thinned-skull window showing AAV expression (red) in the somatosensory cortex (scale bar 1 mm). Right, vasculature within box of A. (**B**) Representative image of AAV-hSYN-HA-hM3D(Gq)-mCherry expressing cortex. Left is wide field image of hSyn-mCherry DREADDs virus (red) in cortex with DAPI (blue) staining (scale bar 100 µm). Right is magnified image of hSyn-mCherry DREADDs virus (red) in cortex with DAPI (blue) and GFAP (green) staining (scale bar 30 µm), showing the virus was not expressed in astrocytes. Data in (**C–E**) are from mice injected with AAV-hSYN-HA-hM4D(Gi)-mCherry, data in (**F–H**) is from mice injected with AAV-hSYN-HA-hM3D(Gq)-mCherry. (**C**) LFP power spectra during stationary periods (basal) and locomotion after CNO injection in hSyn G(i) DREADDs mice, normalized to stationary periods during vehicle injection in the same animal. (**D**) Plot of basal arteriole diameter after vehicle injection (x-axis) versus CNO injection (y-axis). (**E**) Population locomotion-triggered averages for vessels ≤ 25 µm in diameter after vehicle and CNO injections. (**F**) LFP power spectra during stationary periods (basal) and locomotion after CNO injection in hSyn G(q) DREADDs mice, normalized to stationary periods during vehicle injection of the same mouse. (**G**) Plot of basal arteriole diameter after vehicle injection (x-axis) versus CNO injection (y-axis). (**H**) Population locomotion-triggered averages in response to locomotion after vehicle and CNO injections.

The online version of this article includes the following figure supplement(s) for figure 2:

**Figure supplement 1.** No significant effect of CNO on basal arteriole diameters or neural activity.

**Figure supplement 2.** hM4D-G(i) DREADDs cause bursts on neural activity.

response was minimal (*Figure 2E*, locomotion-evoked response: vehicle 20.5 ± 2.0%, CNO 18.3 ± 4.1%, LME p<1, n = 7 mice, 25 arterioles in FL/HL representation).

When we pan-neuronally expressed hM3D-G(q) DREADDs, which will increase excitability and intracellular calcium levels (*Alexander et al., 2009*; *Pati et al., 2019*), activation of the DREADDs

with CNO injections caused increases in gamma-band power relative to vehicle (*Figure 2F*, basal 18.6 ± 11.2%, p<0.02; locomotion, 43.6 ± 15.0%, p<7.1×10$^{-3}$). Injections of CNO caused substantial increases in the basal arterial diameter over controls (*Figure 2G*, +18.8 ± 5.6%, LME p<0.03, n = 7 mice, 37 vessels). The increase in basal arteriole diameter by hM3D-G(q) DREADD activation was comparable to the locomotion-evoked response (*Figure 2H*, locomotion-evoked response: vehicle 18.6 ± 1.7%, CNO 26.0 ± 4.2%, LME p<1, n = 7 mice, 23 arterioles in FL/HL representation). These results suggest that neurons, or a subset of neurons, tonically release vasodilator(s) due to ongoing background neural activity, and elevation of this activity can drive arterial dilations.

## Altering pyramidal neuron excitability drives large neural activity changes without corresponding changes in arterial diameter

The importance of local neural activity on the regulation of arterial dilation raises the question of which sub-population(s) of neurons are involved. We manipulated the excitability of pyramidal neurons by injecting AAVs encoding hM3D(q) or hM4D(i) DREADDs under a CaMKIIa promoter, which will restrict the expression to pyramidal neurons (*López et al., 2016*; *Yuan and Grutzendler, 2016*). CaMKIIa-DREADDs labeled cells colocalized with CaMKIIa expression and were primarily localized to neurons with cell bodies in layer 5 (*Figure 3A*). Previous work has shown that optogenetic stimulation of layer 5 pyramidal neurons drives increases in neural activity in all layers via recurrent connections, and these increases in neural activity are very similar to what is seen with sensory stimulation (*Vazquez et al., 2014*; *Vazquez et al., 2018*), so chemogenetic stimulation of these cells will likely produce neural activity patters similar to those normally found in the cortex. The dilatory signal in the deeper cortical layers is electrically conducted through the vasculature rapidly by gap junctions (*Bagher and Segal, 2011*; *Welsh et al., 2018*) to the pial vessels (*Rungta et al., 2018*). Activating hM4D(i) DREADDs in excitatory neurons with CNO caused large decreases in basal gamma-band power (*Figure 3B and D*, basal −36.0 ± 16.4%, p<0.01; locomotion, −26.8 ± 15.8%, p<0.02). When hM3D(q) DREADD receptors in excitatory neurons were activated, a corresponding large increase in basal gamma-band power was observed (*Figure 3C and G*, basal +52.4 ± 18.9%, p<5.0×10$^{-3}$; locomotion, +50.9 ± 31.6%, p<0.02). Despite the large changes in neural activity, we did not see significant changes in the basal arterial diameters with manipulation of pyramidal neuron activity (−2.2 ± 5.7%, LME p<1, n = 7 mice, 38 arterioles for inhibition and +7.2 ± 8.8%, LME p<0.66, n = 6 mice, 41 arterioles for excitation) (*Figure 3E* and *Figure 3H*, respectively). The locomotion-evoked dilations were also not affected by manipulations of neural excitability (*Figure 3F* G(i) DREADDS: vehicle 25.3 ± 1.9%, CNO 21.6 ± 4.3%,LME p<1, n = 7 mice, 29 arterioles in FL/HL representation; *Figure 3I* G (q) DREADDS: vehicle 19.1 ± 2.3%, CNO 17.3 ± 4.7% LME p<1, n = 6 mice, 13 arterioles in FL/HL representation).

The lack of change in the arterial diameters could not be due to a lack of an effect of the chemogenetic manipulations as the changes in neural activity were very large. The decreases in basal neural activity generated by activation of G(i)-coupled DREADDs in pyramidal neurons were nearly as large as those elicited by muscimol (*Figure 3B*), and the increases in basal neural activity generated by activation of G(q)-coupled DREADDs in pyramidal neurons were as large as those seen during locomotion (*Figure 3C*), yet in neither of these cases did we observe arterial diameter changes of similar magnitude. Thus, pyramidal neuron activity is a strong driver of gamma band LFP power, but only has a moderate effect on basal arterial diameter, and little effect on the locomotion-evoked dilations.

## nNOS-expressing neurons and NO signaling control arterial diameter independent of overall neural activity

Previous work has implicated neuronally-generated nitric oxide (NO) (*Hosford and Gourine, 2019*), generated by interneurons (*Krawchuk et al., 2020*; *Lee et al., 2020*) in the dilation of cerebral arteries. To test the role of neuronal nitric oxide synthase (nNOS) expressing neurons, we expressed hM3D(q) and hM4D(i) DREADDs along with the reporter protein mCherry in nNOS-expressing neurons using flexed AAVs in nNOS-cre- mice (B6.129-*Nos1*$^{tm1(cre)Mgmj}$/J, Jackson Laboratory #017526) (*Leshan et al., 2012*; *Augustine et al., 2018*). Expression of the reporter protein mCherry was primarily in the deeper and more superficial layers of the cortex (*Figure 4A,B*), consistent with the observed densities of nNOS-expressing neurons (*Perrenoud et al., 2012*). Co-labeling with

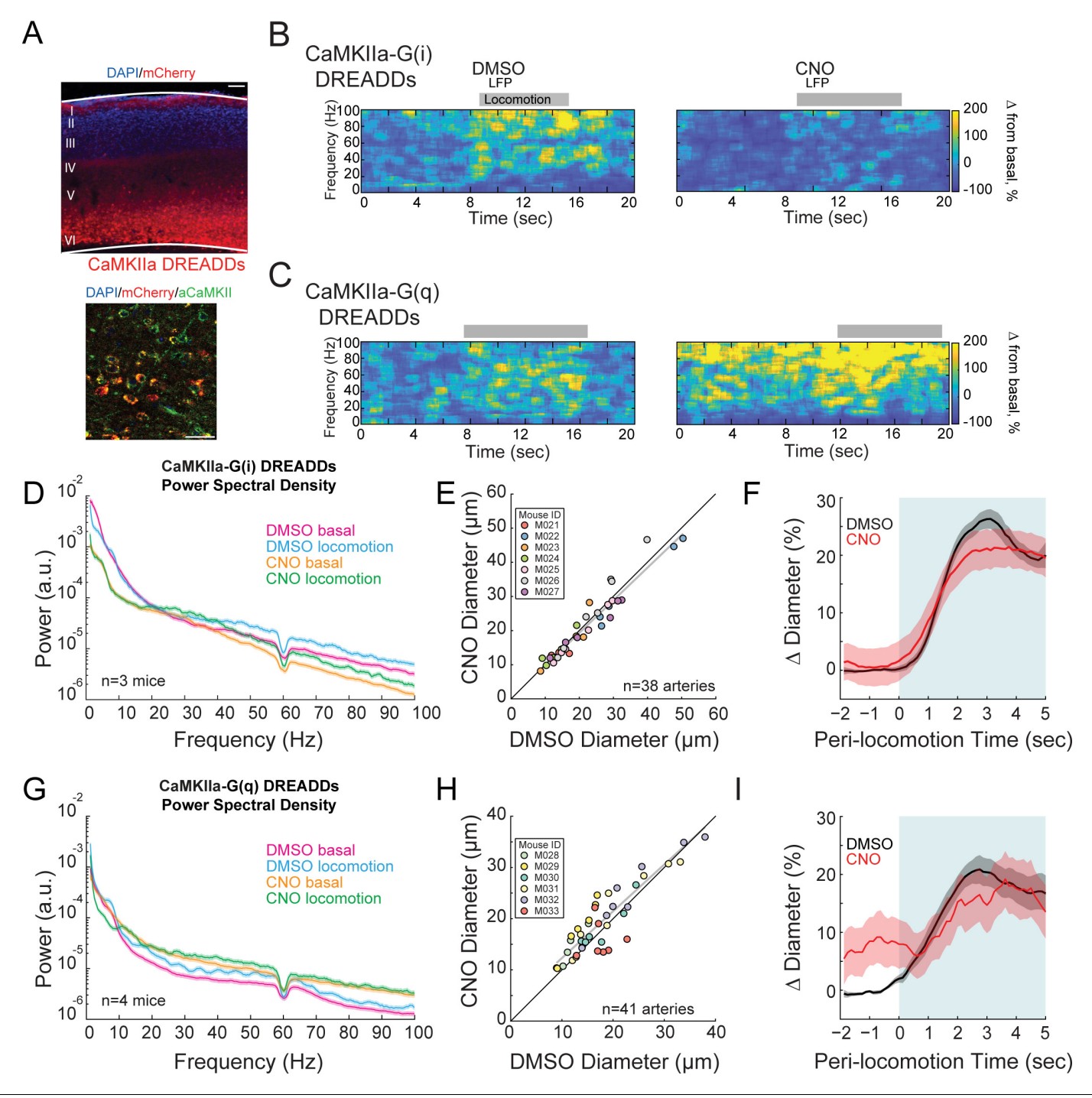

**Figure 3.** Activity, but not of pyramidal neurons, bidirectionally controls basal arteriole diameter. (**A**) Representative image of AAV-CaMKIIa-hM3D(Gq)-mCherry infected cortex, where DREADDs are expressed under a CaMKIIa promoter. Top image is wide field image of CaMKIIa-mCherry DREADDs virus (red) in cortex with DAPI (blue) staining. Scale bar 100 μm. Bottom is a magnified image of CaMKIIa-mCherry DREADDs (red) expressed in cortex with DAPI (blue) and αCaMKII (green) staining showing expression in excitatory neurons. Scale bar 30 μm. (**B**) Representative LFP spectrograms of CaMKIIa-G(i) DREADD expressing mouse after vehicle (left) or CNO (right) injection. Locomotion events are denoted by shading. (**C**) Representative LFP spectrograms of CaMKIIa-G(q) DREADD expressing mouse (expression in pyramidal neurons) after vehicle (left) or CNO (right) injection. Data in **D-F** are from mice injected with AAV-CaMKIIa-hM4D(Gi)-mCherry, data in **G-H** is from mice injected with AAV-CaMKIIa-hM4D(Gq)-mCherry (expression in pyramidal neurons). (**D**) LFP power spectra during stationary periods (basal) and locomotion after CNO injection in CaMKIIa G(i) DREADDs mice, normalized to stationary periods during vehicle injection of the same mouse. (**E**) Plot of basal arteriole diameter after vehicle injection (x-axis) versus CNO injection (y-axis). (**F**) Population locomotion-triggered averages after vehicle and CNO injections for arterioles ≤ 25 μm in basal diameter. (**G**) LFP

*Figure 3 continued on next page*

*Figure 3 continued*

power spectra during stationary (basal) and locomotion periods after CNO injection in CaMKIIa G(q) DREADDs mice, normalized to stationary periods during vehicle injection of the same mouse. (**H**) Plot of basal arteriole diameter after vehicle injection (x-axis) versus CNO injection (y-axis). (**I**) Population locomotion-triggered averages after vehicle and CNO injections.

antibodies for nNOS (*Figure 4A,B*) showed co-expression (white) of mCherry (magenta) and nNOS (green) in both cell bodies and processes of neurons. Neurons showing strong anti-nNOS labeling were taken to be type 1 nNOS neurons, with-type 2 nNOS neurons showing less intense anti-nNOS labeling, consistent with previous anatomical studies (*Perrenoud et al., 2012*). Immunolabeling (n = 2 mice) revealed that 28.6% (2/7 cells) type 1% and 35.2% (19/54) of type 2 nNOS neurons expressed DREADDs, and that 44.6% (21/47) of DREADD-expressing cells were nNOS positive. Neurons expressing mCherry also expressed GAD65, indicating they were GABAergic interneurons (*Figure 4C*).

CNO injections in mice expressing G(i)-DREADD in nNOS-expressing neurons resulted in no significant change in gamma-band power (*Figure 4D*, basal −1.2 ± 11.6%, p<1; locomotion, −5.7 ± 2.2%, p<0.09). Increasing the excitability of nNOS-expressing neurons resulted in a significant decrease in basal gamma-band power (*Figure 4G*, basal −19.8 ± 2.6%, p<$4.0 \times 10^{-3}$; locomotion, −12.8 ± 4.7%, p<0.10), consistent with these neurons being inhibitory. However, decreasing the excitability of nNOS neurons drove substantial and significant decreases in both basal arteriole diameter (−10.3 ± 6.4% LME p<$1.3 \times 10^{-4}$, n = 9 mice, 49 vessels *Figure 4E*) and the locomotion-induced dilation (*Figure 4F*, vehicle 15.3 ± 2.0%, CNO 0.3 ± 4.3%, LME p<$6.3 \times 10^{-3}$, n = 6 mice, 22 arterioles in FL/HL representation). Elevating the excitability of nNOS-expressing neurons significantly increased the basal arterial diameter (+5.8 ± 6.6%, LME p<$2.7 \times 10^{-4}$, n = 8 mice, 52 vessels, *Figure 4H*), but did not increase the locomotion-evoked increase above the control (*Figure 4I* vehicle 17.4 ± 1.8%, CNO 19.6 ± 3.8%, LME p<1, n = 8 mice, 25 arterioles in FL/HL representation). Interestingly, the diameter changes moved in the opposite direction as neural activity when the excitability of nNOS-expressing neurons was increased, consistent with nNOS neurons being inhibitory.

We then tested whether the effects of nNOS-expressing neuron activity on arteriole diameter was mediated by NO, as these neurons can also release vasoactive peptides (*Cauli et al., 2004*). Previous studies with NOS inhibitors have shown conflicting effects (*Lindauer et al., 1999*; *Stefanovic et al., 2007*). Discrepancy in the literature could be due to differences in experimental methodology: topical application of drugs may not reach lower cortical layers (*Greenberg et al., 1997*; *Ferezou et al., 2006*) where most of the highly expressing nNOS neurons reside, and systemic administration impacts the cardiovascular system. To avoid these experimental confounds, we infused the water-soluble NO synthase inhibitor, Nω-nitro-L-arginine methyl ester (L-NAME) through an implanted cannula to decrease NO-mediated signaling. In our hands, pharmacological infusions through cannula affect all cortical layers within ~1.5 mm of the infusion site (*Winder et al., 2017*). Note that recent studies, using both single-cell genomic methods as well as immunohistochemistry, have shown that some cortical neurons express eNOS (*Yousef et al., 2004*; *Lein et al., 2007*; *Tasic et al., 2016*), and that some endothelial cells of cerebral arterioles express nNOS (*Vanlandewijck et al., 2018*; *Zeisel et al., 2018*), so the endothelial/neural specificity of these enzymes is not complete. Additionally, crystal structures and careful pharmacology experiments have revealed that 'selective' pharmacological inhibitors of nNOS are much less selective than was originally thought (*Bland-Ward and Moore, 1995*; *Reiner and Zagvazdin, 1998*; *Engelhardt et al., 2006*; *Pigott et al., 2013*; *Poulos and Li, 2017*), so it is likely that there are no inhibitors of nNOS or eNOS are specific enough to target one enzyme over another in vivo. Finally, locomotion-evoked dilations are unlikely to be mediated by eNOS expressed in endothelial cells as increases in eNOS activity in response to flow-evoked changes in diameter evolve over minutes, not seconds (*Kim et al., 2016*) (too slow to account for the locomotion-evoked dilations), and there are no shear-stress changes seen with sensory-evoked dilation (*Ngai and Winn, 1996*). Blocking the production of NO caused no significant change in the basal gamma-band power (*Figure 4—figure supplement 1A*, basal −5.0 ± 8.5%, p<0.55; locomotion, −7.5 ± 9.6%, p<0.43), but drove a decrease in basal and locomotion-evoked arterial diameter (−10.6 ± 4.6%, LME p<$7.6 \times 10^{-6}$, n = 6 mice, 36 arterioles, and vehicle 13.2 ± 2.4%, L-NAME 4.2 ± 3.8%, LME p<0.04, n = 6 mice, 20 arterioles in FL/HL representation, *Figure 4—*

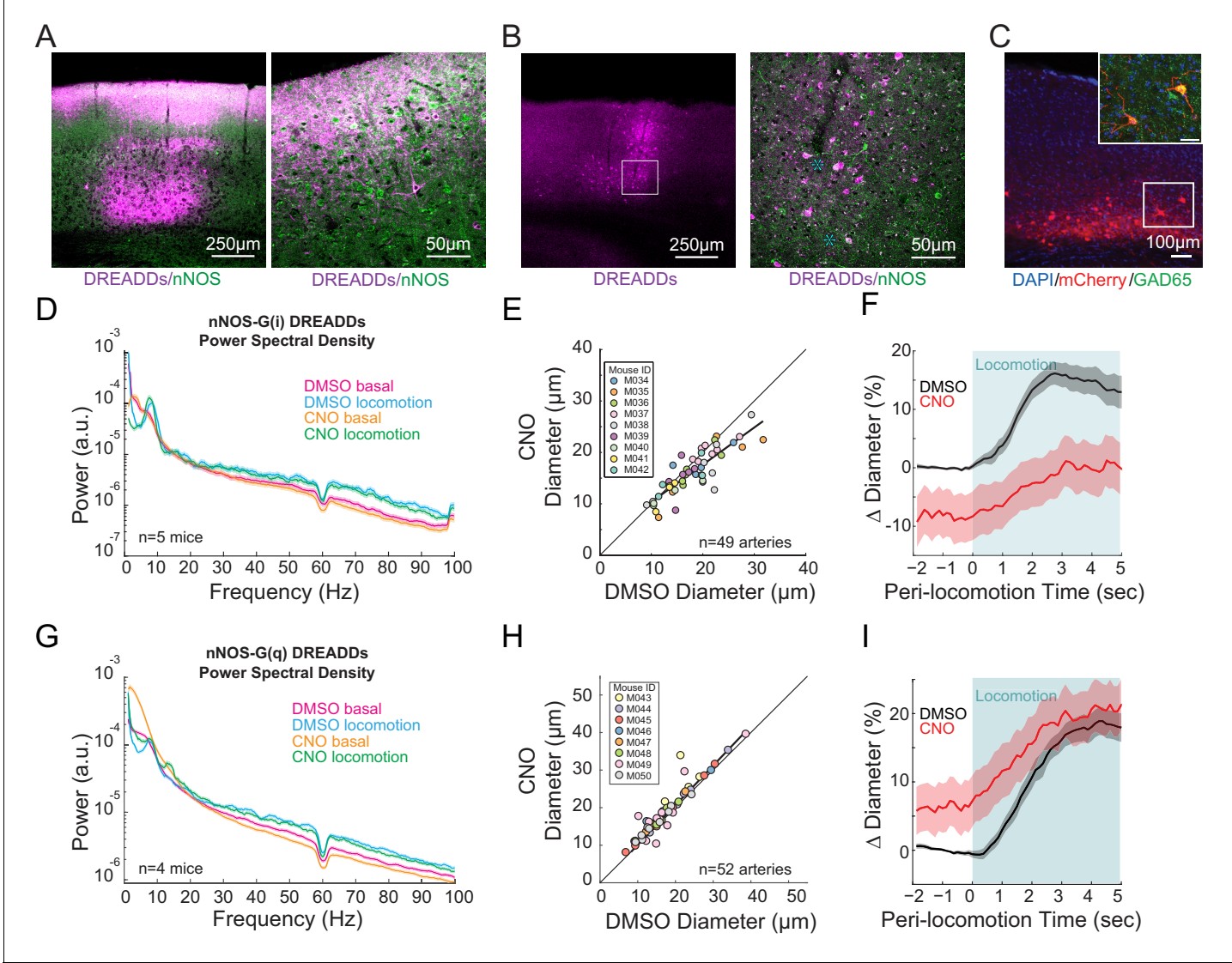

**Figure 4.** nNOS expressing neurons controls arteriole diameter independent of overall neural activity. (**A**) Representative image of cortex taken of AAV-hSyn-DIO-hM3D(Gq)-mCherry in nNOS-cre mice, where DREADDs expressed in nNOS+ cells. The mCherry label is magenta, nNOS antibody is green. Co-labeling in processes in cell bodies shows up as white. Blue asterixis denote putative type 1 nNOS neurons. Dark vertical streaks are blood vessels. (**B**) Left, wide-field image of DREADD-infected somatosensory cortex. Right, enlargement of area in the white box, showing expression around blood vessel. (**C**) Left image is wide-field image of nNOS-mCherry DREADDs expression (red) in cortex with DAPI (blue) staining. Scale bar 100 µm. Right image is zoomed image of nNOS-mCherry DREADDs virus (red) in cortex with DAPI (blue) and GAD65 (green) staining, Scale bar 30 µm. Data in **D-F** are from nNOS-cre mice injected with AAV-hSyn-DIO-hM4D(Gi)-mCherry, data in **G-I** is from nNOS-cre injected with AAV-hSyn-DIO-hM4D(Gq)-mCherry. (**D**) LFP power spectra during stationary periods (basal) and locomotion after CNO injection in nNOS G(i) DREADDs mice, normalized to vehicle injection of the same mouse. (**E**) Plot of basal arteriole diameter after vehicle injection (x-axis) versus CNO injection (y-axis). (**F**) Population locomotion-triggered diameter averages after vehicle and CNO injections for arterioles ≤ 25 µm in basal diameter. (**G**) LFP power spectra during stationary periods (basal) and locomotion after CNO injection in nNOS G(q) DREADDs mice. (**H**) Plot of basal arteriole diameter after vehicle injection (x-axis) versus CNO injection (y-axis). (**I**) Population locomotion-triggered averages after vehicle and CNO injections for arterioles ≤ 25 µm in basal diameter.

The online version of this article includes the following figure supplement(s) for figure 4:

**Figure supplement 1.** NO produced by nNOS expressing neurons controls arteriole diameter independent of overall neural activity.

*figure supplement 1B and C*). The basal arterial diameter constrictions produced by L-NAME were not significantly different from those produced by G(i) DREADDs in nNOS neurons (−10.6% vs. −10.3%, LME p<0.57 *Figure 4—figure supplement 1D*). Although there is likely some contribution from the inhibition of eNOS contributing to these constrictions (*Atochin and Huang, 2011*), these results are consistent with the hypothesis that the NO produced by nNOS-expressing neurons contributes maintaining the diameter of cerebral arterioles.

## Pharmacological manipulations of type 1 nNOS neuron activity

As we saw expression of DREADDs primarily in type 2 nNOS neurons, we wondered how activation of type 1 nNOS neurons might affect arteriole dynamics. Type 1 nNOS neurons are unique among cortical neurons in that they express the receptor for Substance P (NK1R) (*Vruwink et al., 2001*; *Tomioka et al., 2005*; *Tomioka and Rockland, 2007*; *Gerashchenko et al., 2008*; *Kilduff et al., 2011*; *Kubota et al., 2011*; *Dittrich et al., 2012*; *Dittrich et al., 2015*; *Endo et al., 2016*; *Williams et al., 2019*; *Williams et al., 2018*), whose activation drives depolarization and spiking (*Dittrich et al., 2012*; *Endo et al., 2016*). As multiple single-cell genomics studies have shown there is no NK1R expressed in mouse cerebral vascular cells or in glial cells (*Vanlandewijck et al., 2018*; *Zeisel et al., 2018*), we can pharmacologically manipulate the activity of Type 1 nNOS neurons independently of other neurons by infusing SP, or an antagonist for the SP receptor, CP-99994 through a chronically implanted cannula. Infusions of CP-99994 caused non-significant changes in gamma-band power (*Figure 5A*, basal −5.4 ± 4.7%, p<0.28; locomotion, +7.3 ± 8.0%, p<0.37) and a non-significant decrease in basal arteriole diameter (−8.0 ± 7.2%, LME p<1, n = 7 mice, 28 vessels, *Figure 5B*) and locomotion-evoked dilation response (vehicle 13.7 ± 2.6%, CP-99994 9.7 ± 3.3%, LME p<1, n = 7 mice, 14 arterioles in FL/HL representation, *Figure 5C*). The lack of a substantial effect suggests that either basal activation of the Substance P receptors in the cortex is low, generating a floor effect and/or the basal activity of the type 1 nNOS neurons is low. When we activated the type 1 nNOS neurons via infusions of SP, we observed non-significant decreases in gamma-band power (*Figure 5D*, basal −20.0 ± 32.4%, p<0.53; locomotion, +5.2 ± 16.0%, p<0.73), but a substantial increase in basal diameter (+13.4 ± 4.4%, LME p<$3.6\times10^{-3}$, n = 7 mice, 30 arterioles, *Figure 5E*), although the effect was not significant for locomotion-evoked responses (vehicle 13.0 ± 2.2%, SP 24.8 ± 4.7%, LME p<0.75, n = 7 mice, 14 arterioles in FL/HL representation, *Figure 5F*). To test the hypothesis that the Substance P facilitated dilation was mediated by increases in neural activity, we co-infused SP with muscimol to block local neural activity, and observed a reduction of the arteriole diameter (−20.0 ± 14.9%, LME p<$3.4\times10^{-7}$, n = 5 mice, 27 arterioles, *Figure 5G*) and no locomotion-evoked dilation (vehicle 14.2 ± 3.0%, SP+muscimol −14.5 ± 3.7%, LME p<$2.8\times10^{-3}$, n = 5 mice, seven arterioles in FL/HL representation, *Figure 5H*), consistent with the action of SP being mediated through local neural activity. These results suggest that type 1 nNOS neurons play an important role in NO-mediated changes in arteriole diameter.

## Role of astrocytes and inwardly rectifying potassium channels

In addition to nitric oxide, there are multiple vasodilatory pathways that have been implicated in signaling the vasculature (*Attwell et al., 2010*; *Otsu et al., 2015*; *Bazargani and Attwell, 2016*; *Drew, 2019*). We first examined how chemogenetic modulation of astrocytes impacted basal and evoked arterial dilations (*Figure 5—figure supplement 1*). We found no significant modulation of arteriole diameter when either G(i)- or G(q)-coupled DREADDs on astrocytes were activated, consistent with previous reports that did not see coupling between astrocytes and evoked arterial responses (*Nizar et al., 2013*; *Bonder and McCarthy, 2014*; *Tran et al., 2018*). This does not rule out an astrocytic role for the control of the cerebral vasculature, as the signaling could work via mechanisms that are not coupled to G-protein signaling.

Recent reports have implicated inwardly-rectifying potassium channels in the capillary bed in mediating functional hyperemia (*Longden et al., 2017*). To test the role of these channels in basal and evoked arterial diameter control, we infused $Ba^{2+}$ or ML-133 (*Longden and Nelson, 2015*), blockers of inwardly rectifying potassium channels. We found that both of these inwardly rectifying $K^+$ channel blockers drove very large increases in neural activity ($Ba^{2+}$, *Figure 5—figure supplement 2*: basal +88.3 ± 44.5%, p<0.01; locomotion, +145.6 ± 86.1%, p<0.02; and ML-133 S12E: basal +147.2 ± 77.7%, p<0.01; locomotion, +157.7 ± 86.4%, p<0.01). These observations of large increase

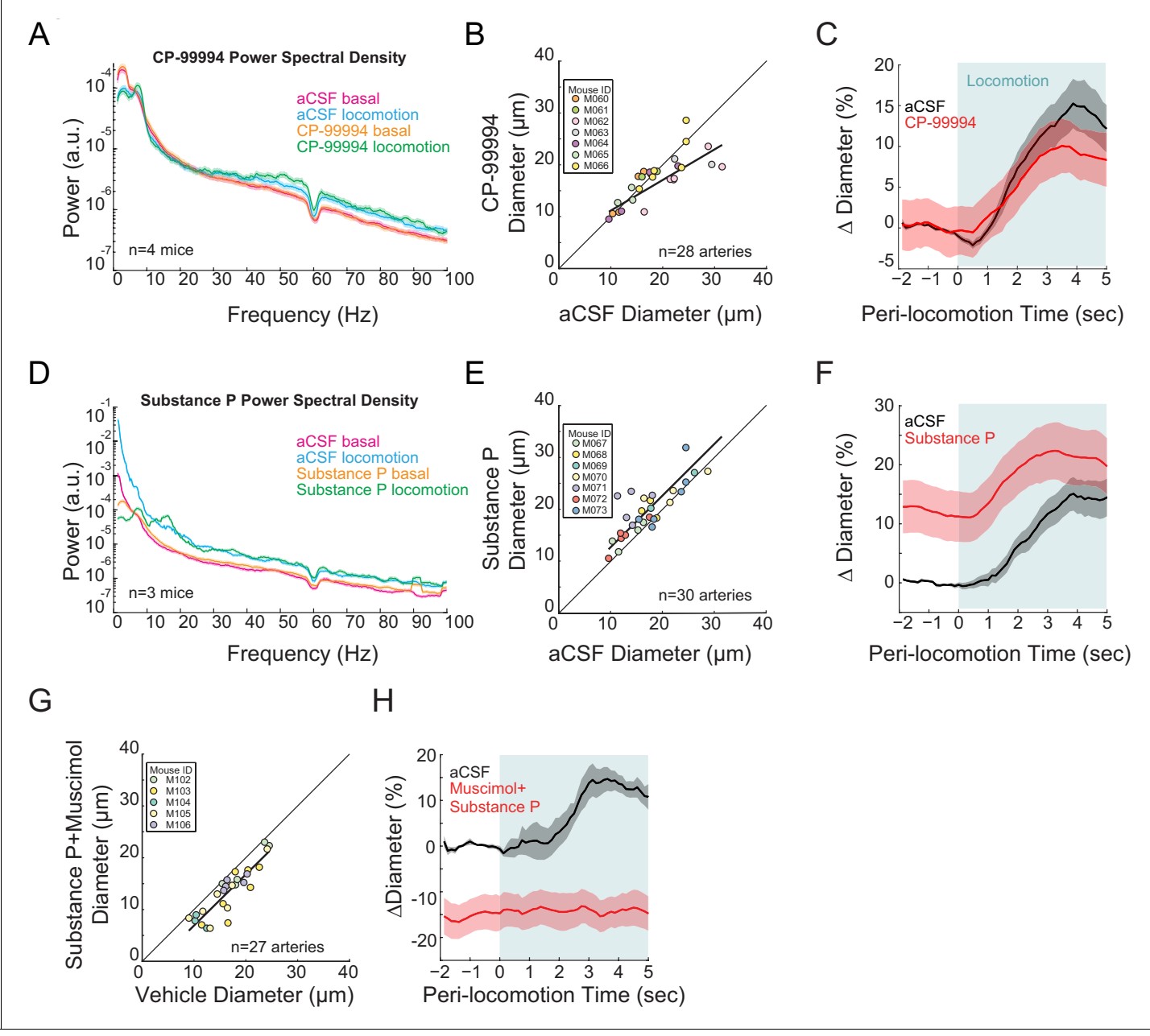

**Figure 5.** Type I nNOS expressing neurons control basal arteriole diameter. Data in **A-C** are from mice after local CP-99994 infusions, which will block the excitatory Substance P receptors on type1 nNOS neurons, data in **D-E** is from mice after Substance P infusions, which will excite Type1 nNOS neurons. (**A**) LFP power spectra during stationary (basal) and locomotion periods after CP-99994 infusion, normalized to vehicle infusion in the same mouse. (**B**) Plot of basal arteriole diameter after vehicle (x-axis) versus CP-99994 infusion (y-axis). (**C**) Population locomotion-triggered averages after vehicle or CP-99994 infusions for arterioles ≤ 25 μm in basal diameter. (**D**) LFP power spectra during stationary (basal) and locomotion periods after Substance P infusion, normalized to vehicle infusion in the same mouse. (**E**) Plot of basal arteriole diameter after vehicle infusion (x-axis) versus Substance P infusion (y-axis). (**F**) Population locomotion-triggered averages after vehicle or Substance P infusions for arterioles ≤ 25 μm in basal diameter. (**G**) Plot of basal arteriole diameter after vehicle infusion (x-axis) versus muscimol and Substance P infusion (y-axis). (**H**) Locomotion-triggered averages after vehicle or muscimol/Substance P infusions for arterioles ≤ 25 μm in basal diameter.

The online version of this article includes the following figure supplement(s) for figure 5:

**Figure supplement 1.** No significant effect of chemogenetic manipulation of astrocytes on basal vessel diameter or neural activity.

**Figure supplement 2.** Kir-channel blockers increase in neural activity and basal arterial diameter.

in basal and evoked neural activity are consistent with previous work that has shown that barium increases the activity of neurons in the awake cortex (*Harnett et al., 2013*). Barium is known to block inwardly rectifying potassium channels found on neurons, such as KIR2.1 (*Alagem et al., 2001*), Kir2.2 (*Amarillo et al., 2014*), and Kir2.3 (*Schram et al., 2003*). Blocking these channels will increase neural excitability, so it is not surprising that barium infusions increase overall neural activity. Blocking inwardly rectifying $K^+$ channels drove basal arteriole dilation (*Figure 5—figure supplement 2*: +12.4 ± 17.2%, LME p<$2.0 \times 10^{-3}$, n = 5 mice, 24 arterioles; S12F: +11.8 ± 17.0% LME p<$3.7 \times 10^{-4}$, n = 5 mice, 36 arterioles), and this basal dilation partially occluded the locomotion-induced response (*Figure 5—figure supplement 2*: vehicle 15.1 ± 2.7%, BaCl$_2$ +20.2 ± 5.3%, LME p<1, n = 5 mice, 16 arterioles in FL/HL representation and *Figure 5—figure supplement 2*: vehicle 14.01 ± 3.0%, ML-133 +21.5 ± 5.8%, LME p<1, n = 5 mice, 26 arterioles in FL/HL representation). As with chemogenetic increases in pan-neuronal activity (*Figure 2*), the increase in neural activity by barium drove a baseline dilation that nearly occluded the locomotion-induced response.

Because of the large increase in neural activity caused by blockade of inwardly rectifying potassium channels in the awake brain, and the likely release of vasodilator resulting from this increase in activity, it is hard to parse out any effects mediated by blockade of inward rectifying channels on the vasculature without transgenic models where the inwardly rectified channels are selectively removed from the vasculature (*Longden et al., 2017*). These results, along with the lack of effects of manipulation of G-protein-coupled pathways in astrocytes on arterial diameter highlight the need for the development of innovative manipulations of signaling pathways in astrocytes and endothelial cells as has been so successfully used for neurons.

## Comparison of pial and penetrating arteriole basal and evoked diameter changes

It is informative to compare the responses of pial and penetrating arterioles to the manipulations performed here. We saw some differences between pial and penetrating arterioles, both in changes to their basal diameters, and in locomotion-evoked dilations. When we looked at basal diameter changes, we saw a significantly smaller constriction of penetrating arterioles than pial arterioles with muscimol infusions (*Figure 6A*). With pan-neuronal excitatory DREADDs, we saw a significantly smaller dilation in penetrating arterioles than in the pial arteries (*Figure 6D*). For all other manipulations, which in general produced smaller changes than muscimol and pan-neuronal excitatory DREADDs, there was no significant difference in the induced basal diameter changes between penetrating and pial arteries.

We then examined the percentage change in diameter during locomotion in pial and penetrating arterioles, here defined as the change from the respective pre-locomotion baseline normalized by the pre-locomotion baseline, for each of the manipulations. For this metric, a manipulation that causes a 10% constriction of baseline and shows a dilation of 5% above the control baseline will have a 15% locomotion-induced dilation (*Figure 6—figure supplement 1*). If the vessel has a 10% baseline dilation and during locomotion dilates to 15% above the control baseline, the percentage change will be 5%. This metric only captures the locomotion-induced dilation component and differs from the measures of dilation reported in the sections above, which included the effects of the baseline shift in diameter. When we looked at all the control condition (vehicle injections and aCSF infusions *Figure 6—figure supplement 2*), we saw a large and substantial difference in the pial arterial dilation versus the penetrating arteriole dilation (17.57 ± 9.35% pial vs. 6.79 ± 6.54% penetrating, LME p<$8 \times 10^{-37}$). This difference is consistent with previous measurements of pial and penetrating arteriole dynamics during locomotion (*Gao et al., 2015*), so we treated penetrating and pial arteries as separate groups for further statistical analysis. We then compared the percentage diameter change during locomotion for pial arterioles and penetrating arterioles (*Figure 6—figure supplement 2*). With this metric, we saw significant differences in the locomotion-evoked responses of pial arterioles with muscimol, pan-neuronal excitation (hsyn-Gq), excitation of pyramidal neurons (CaMKIIa-Gq), inhibition of nNOS neurons (nNOS-Gi) and L-NAME. Interestingly, the expression of excitatory DREADDs in pyramidal neurons and pan-neuronally had the paradoxical effect of decreasing the amplitude of the locomotion-evoked dilation when quantified relative to its within treatment baseline. This result can be explained by the increased baseline neuronal activity increasing baseline arteriole diameters while not extending the maximum amplitude of arterial dilations. However, none

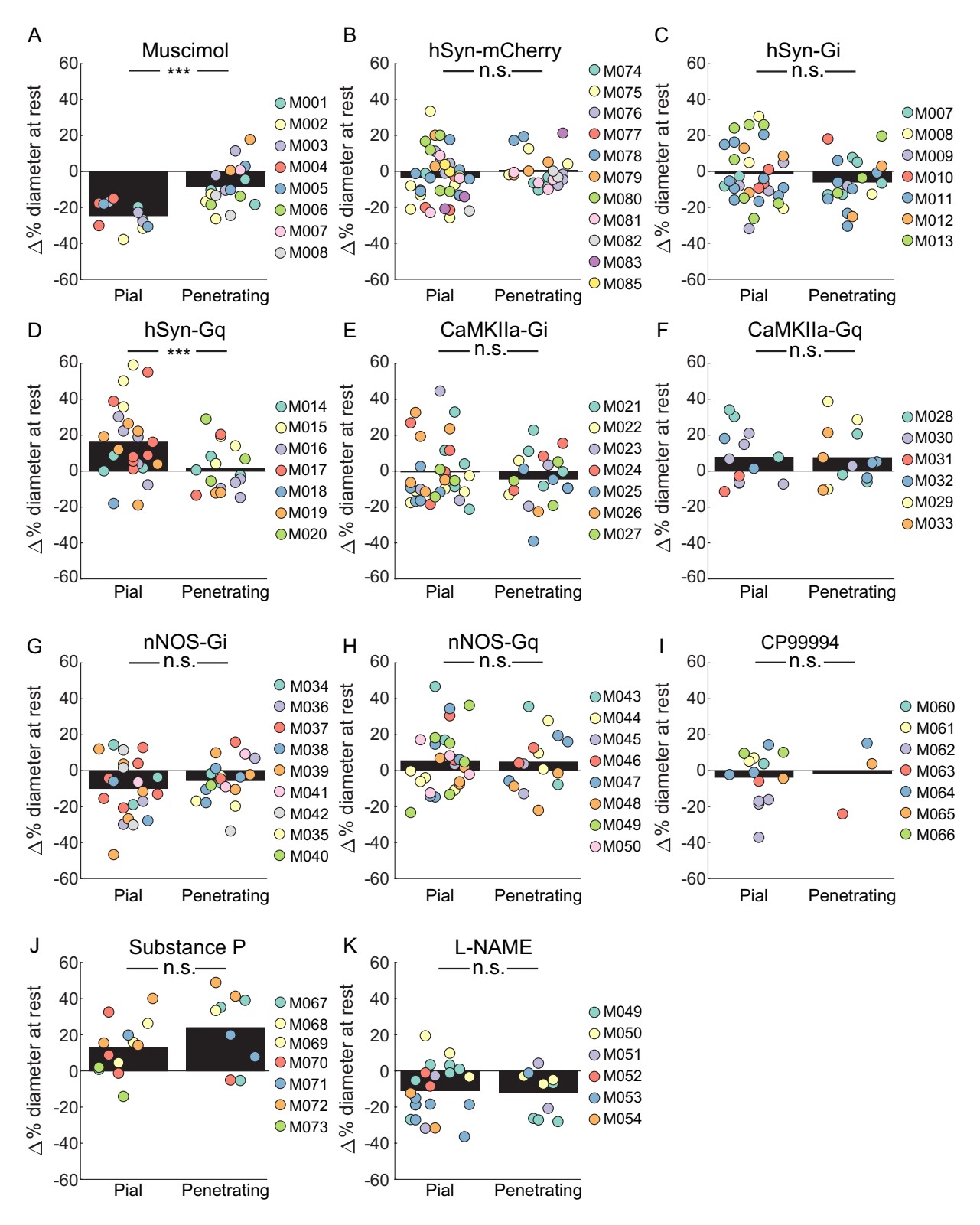

**Figure 6.** Comparisons of the effects of various manipulations on the basal diameters of pial and penetrating arterioles. (**A**) Comparison of change in resting vessel diameter in penetrating and pial arteries after Muscimol infusion. There was a significant difference in the change of baseline diameter between pial and penetrating arteries with pial arteries displaying a larger change. (LME $p < 1.5 \times 10{-6}$, n = 8 mice, 12 pial, 18 penetrating vessels). (**B**) Comparison of change in resting vessel diameter in penetrating and pial arteries in hSyn-mCherry-expressing animals after CNO infusion. No significant

*Figure 6 continued on next page*

*Figure 6 continued*

difference was observed in change in baseline diameter between pial and penetrating arteries (LME p<0.25, n = 11 mice, 40 pial, 22 penetrating vessels). (C) Comparison of change in resting vessel diameter in penetrating and pial arteries in hSyn-Gi expressing animals after CNO infusion. No significant difference was observed in change in baseline diameter between pial and penetrating arteries. (LME p<0.3, n = 7 mice, 33 pial, 20 penetrating vessels). (D) Comparison of change in resting vessel diameter in penetrating and pial arteries in hSyn-Gq expressing animals after CNO infusion. There was a significant difference between the change in baseline diameter between pial and penetrating arteries with pial arteries displaying a larger change (LME p<0.001, n = 7 mice, 25 pial, 17 penetrating vessels). (E) Comparison of change in resting vessel diameter in penetrating and pial arteries in CaMKIIa-Gi expressing animals after CNO infusion. No significant difference was observed in change in baseline diameter between pial and penetrating arteries. (LME p<0.5, n = 7 mice, 33 pial, 20 penetrating vessels). (F) Comparison of change in resting vessel diameter in penetrating and pial arteries in CaMKIIa-Gq expressing animals after CNO infusion. No significant difference was observed in change in baseline diameter between pial and penetrating arteries. (LME p<1, n = 6 mice, 13 pial, 13 penetrating vessels). (G) Comparison of change in resting vessel diameter in penetrating and pial arteries in nNOS-Gi expressing animals after CNO infusion. No significant difference was observed in change in baseline diameter between pial and penetrating arteries. (LME p<0.3, n = 9 mice, 22 pial, 19 penetrating vessels). (H) Comparison of change in resting vessel diameter in penetrating and pial arteries in nNOS-Gq expressing animals after CNO infusion. No significant difference was observed in change in baseline diameter between pial and penetrating arteries. (LME p<1, n = 8 mice, 28 pial, 15 penetrating vessels). (I) Comparison of change in resting vessel diameter in penetrating and pial arteries after CP99994 infusion. No significant difference was observed in change in baseline diameter between pial and penetrating arteries (LME p<1, n = 7 mice, 14 pial, three penetrating vessels). (J) Comparison of change in resting vessel diameter in penetrating and pial arteries after Substance P infusion. No significant difference was observed in change in baseline diameter between pial and penetrating arteries (LME p<0.15, n = 7 mice, 13 pial, 9 penetrating vessels). (K) Comparison of change in resting vessel diameter in penetrating and pial arteries after L-NAME infusion. No significant difference was observed in change in baseline diameter between pial and penetrating arteries (LME p<0.5, n = 6 mice, 22 pial, 10 penetrating vessels).

The online version of this article includes the following figure supplement(s) for figure 6:

**Figure supplement 1.** Schematic showing the calculation of the locomotion-evoked component of the diameter change.
**Figure supplement 2.** Locomotion triggered average diameter changes relative to locomotion diameter.

of our manipulations produced a significant change in the locomotion-evoked diameter change of penetrating arterioles from the control condition (*Figure 6—figure supplement 2*).

Overall, the locomotion-evoked changes in penetrating arteriole diameters were smaller than those in the pial arteries, and the size of the penetrating arteriole dilations were not affected by our manipulations. The difference observed between penetrating and pial arterioles cannot be due to imaging artifacts, as sub-micron cellular structures like spines and microglial processes can be imaged at this depth with a thinned skull window (*Drew et al., 2010b*; *Shih et al., 2012a*), and our diameter measurement algorithm (see Materials and methods) has been validated to work robustly in much lower signal-to-noise regimes than present here (*Gao and Drew, 2014*). There are several non-exclusive possibilities for the differences between the responses of pial and penetrating arteries to behavioral and experimenter-evoked changes in neural activity. One possible reason for this difference could be that penetrating arterioles are less sensitive to vasodilators, with a less steep voltage-diameter relationship that pial arterioles. As there are differences in mural cell morphology at different levels of the arteriole tree (*Hartmann et al., 2015*), it is possible that there are some previously unobserved differences in the smooth muscle or endothelial cells between penetrating and pial arteries. When mice are anesthetized with the powerful vasodilator isoflurane, pial and penetrating arteries dilate to the same extent, suggesting that the vessels have the same maximal dilation (*Gao et al., 2015*). The diameter of arterioles is linearly related to the membrane potential of smooth muscle cells (*Wölfle et al., 2011*), and any electrical signals at one point are conducted via gap junctions up to several hundred microns away (*Segal, 2005*; *Bagher and Segal, 2011*; *Hald et al., 2012*). As the pial arteries may be further up the arterial tree than penetrating arteries, they will be also integrating signals from many more smaller vessels (*Segal, 2000*; *Welsh et al., 2018*), and this may account for the differences in the amplitudes of the responses of the two vessel types. Another possibility could be the spatial location of the generation of the vasodilator(s) relative to the arterioles, particularly in the case of nitric oxide. Simulation of NO diffusion and degradation dynamics in the tissue have shown that NO production needs to be very close (of order microns to tens of microns) to the arteriole in order to exert an appreciable dilatory effect (*Haselden et al., 2020*). It could be that NO is produced closer to pial vessel (potentially in the processes of more superficial type two neurons), than near the penetrating arterioles in the superficial layers. Finally, the different mechanical environments of the pial and penetrating arterioles may account for their differing responses. The pial arterioles are surrounded by CSF (*Abbott et al., 2018*), whereas the

penetrating arterioles are surrounded by a thin paravascular space (*Iliff et al., 2012*) filled with collagen fibers tethering the artery to the brain (*Roggendorf and Cervós-Navarro, 1977*). When the smooth muscle cells surround pial arteries relax, the vessel's dilation need only displace CSF, which has viscosity near that of water, while dilation (or constriction) of penetrating arterioles must displace (or pull upon) the viscoelastic brain tissue (*Goriely et al., 2015*; *Weickenmeier et al., 2018a*; *Weickenmeier et al., 2018b*; *Kedarasetti et al., 2020*). The viscoelasticity of brain tissue will function as an ultra-low-pass mechanical filter on diameter changes of penetrating arterioles, reducing the amplitude of 'rapid' dilations (on the order of seconds), although allowing slower dynamics (on the order of minutes), which could explain why we saw similar changes in both pial and penetrating arterioles baseline (which take place over minutes), but no effects on the 'rapid' locomotion-evoked dilations. Studies in peripheral vascular structures have shown that when blood vessels are imbedded in tissue, the mechanics of the surround tissues dominates and reduces their dilation (*Fung et al., 1966*; *Liu et al., 2007*). Brain tissue is stiffer when the skull is intact than after a craniotomy (*Hatashita and Hoff, 1987*). Brain tissue exhibits strain-stiffening (becoming stiffer under pressure) (*Marmarou et al., 1975*; *Marmarou et al., 1978*), and locomotion is accompanied by an elevation of intracranial pressure (*Gao and Drew, 2016*; *Norwood et al., 2019*; *Eftekhari et al., 2020*). All these factors may exacerbate mechanical restriction of dilation by stiffening the brain tissue, preventing full expression of the arterial dilation to locomotion in penetrating arterioles. These results show that pial and penetrating arterioles display similar basal diameter changes for manipulations that produced moderate changes in diameter, although the locomotion-evoked changes in diameter of penetrating arterioles were not affected by any of the manipulations of neural activity. Elucidating the vascular, neural, and/or mechanical origins of the differences in the responses of pial and penetrating arterioles will be informative for understanding their roles in regulating cerebral blood flow (*Lorthois et al., 2011*; *Gould et al., 2017*; *Schmid et al., 2017*).

## Locomotion drives robust Ca$^{2+}$ responses in nNOS neurons

If nNOS neurons contribute to the locomotion-induced dilation of arteries, then we would expect their activity to increase during locomotion. Electrophysiological recordings have shown that voluntary locomotion drives large increases in the activity of pyramidal neurons and interneurons in FL/HL representation (*Figure 1—figure supplement 1*; *Zhang et al., 2019*), but it is not known if nNOS-positive interneurons exhibit increased activity, as their activity cannot be separated out electrophysiologically. To determine how nNOS-positive interneuron activity is affected by locomotion, we used fiber photometry (*Schulz et al., 2012*) to simultaneously monitor the activity of nNOS neurons expressing GCaMP6s (*Chen et al., 2013*) and local cerebral blood volume deep in the parenchyma, which was labeled with an intravenous injection of TRITC dextran (*Cerri et al., 2019*). Calcium signals have been shown to be strong quantitative predictors of hemodynamic signals (*Aydin et al., 2020*). Fiber photometry will record bulk calcium signals from the processes and cell bodies of a specific neuronal cell type, and these calcium signals should correlate with nitric oxide production (whose synthesis is known to be calcium dependent) in nNOS-expressing neurons. Fiber photometry is not imaging forming, and will report the summed fluorescence of neurons within ~200 μm from the tip of the fiber (*Pisanello et al., 2019*), so the contributions of type 1 and type 2 nNOS neurons cannot be differentiated. However, given the low density of type 1 nNOS neurons, the signal is likely to be primarily from type 2 nNOS neurons. We corrected for the absorbance of fluorescence by hemoglobin (*Gunaydin et al., 2014*; *Cerri et al., 2019*) in the blood using the TRITC signal, and validated this method on mice that expressed GFP ubiquitously (*Okabe et al., 1997*; *Figure 7—figure supplement 1*). The slow (on the timescale of tens to hundreds of seconds) rise and fall of the CBV signals is due to the venous component. The venous contribution to the CBV signal is known to be larger than the arterial component (*Kim et al., 2007*; *Huo et al., 2015b*; *Huo et al., 2015a*) and have slower dynamics than the arterial component (*Kim et al., 2007*; *Drew et al., 2011*; *Moon et al., 2013*; *Gao et al., 2015*; *Huo et al., 2015b*; *Huo et al., 2015a*; *Gao and Drew, 2016*), particularly in the parenchyma (*Kim and Kim, 2010*). We found that calcium signals from GCaMP expressed pan-neuronally, in pyramidal neurons, and in nNOS-positive neurons greatly increased during locomotion (*Figure 7A–C,E* Population GCaMP intensity change in z-units: hSyn: 2.51 ± 0.15, LME difference from CaMKII: p<0.44; CaMKII: 2.59 ± 0.38; nNOS: 1.94 ± 0.37, LME difference from CaMKII: p<1e-3). Activity in all these neuron types was correlated with vasodilation (*Figure 7—figure supplement 2*, CaMKII: 0.60 ± 0.17; hSyn: 0.54 ± 0.14, LME difference from CaMKII: p<0.7;

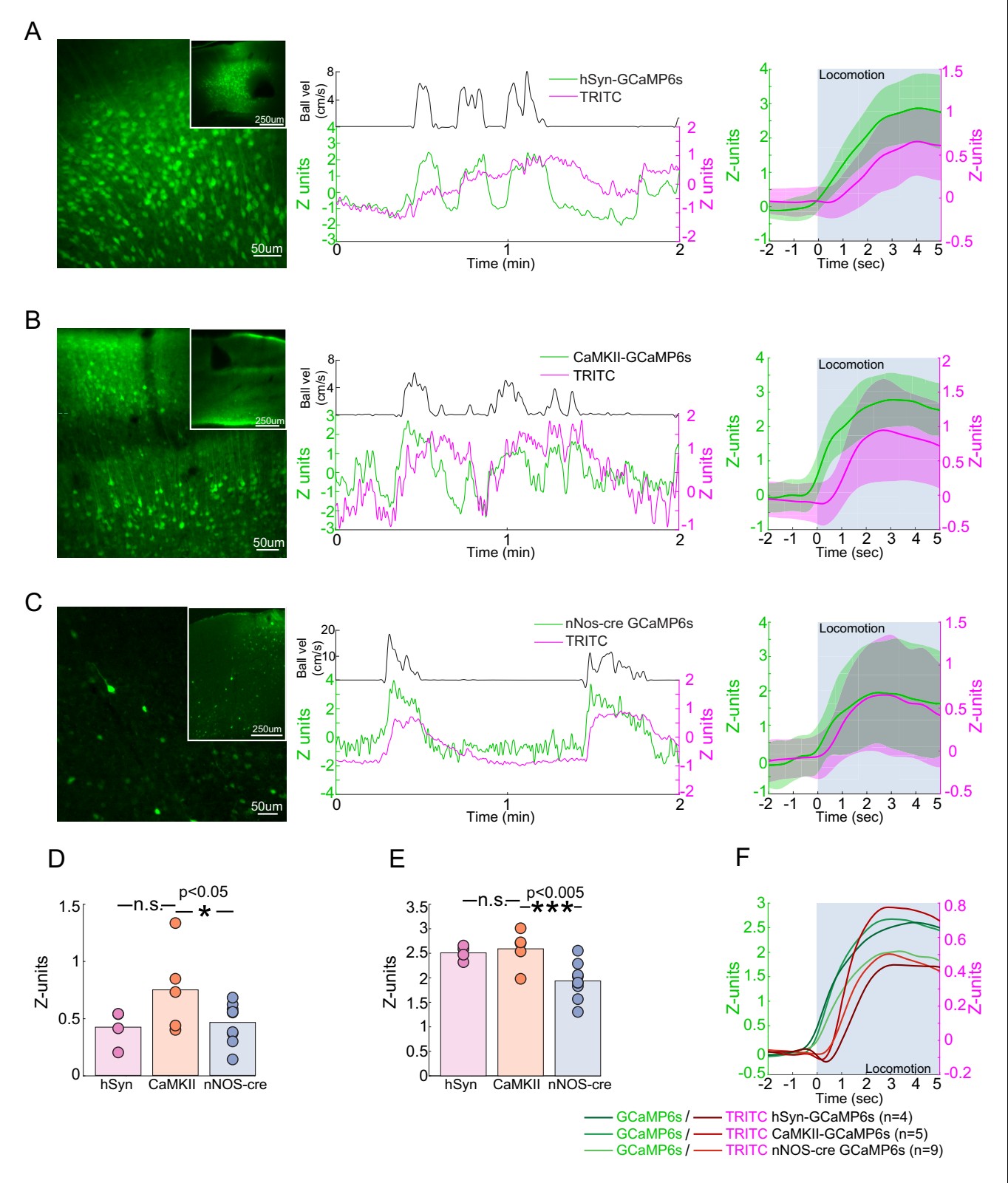

**Figure 7.** Locomotion-evoked Ca²⁺ signals in neurons. (A–C) Left, a representative image of cortex taken of AAV-hSyn-GCaMP6s in C57BL/6J mice (**A**), AAV-CaMKII-GCaMP6s in C57BL/6J mice (**B**) and AAV-Syn-FLEX-GCaMP6s in nNOS-cre mice. Note that there are substantially fewer cells labeled in AAV-Syn-FLEX-GCaMP6s in nNOS-cre mice. Center, a representative trace of TRITC (blood volume - magenta) and GCaMP6s (green) changes evoked by a single locomotion event (black trace above). Right, average locomotion-evoked blood volume and GCaMP6s changes of events >5 s and <10 s in

*Figure 7 continued on next page*

*Figure 7 continued*

duration from the example animal. Locomotion period shaded in blue. (D) Population average locomotion-evoked blood volume change. Bars are population mean of each GCaMP6s sub-type; circles are individual animal averages. (E) Population average locomotion-evoked GCaMP6s fluorescence change. Bars are population mean of each GCaMP6s sub-type; circles are individual animal averages. (F) Population average trace of locomotion-evoked changes in blood volume (red) and GCaMP (green) fluorescence. Period of locomotion is shaded in blue.

The online version of this article includes the following figure supplement(s) for figure 7:

**Figure supplement 1.** Locomotion-evoked $Ca^{2+}$ signals in nNOS neurons, correction of photobleaching and hemodynamic attenuation of GCaMP signals.

**Figure supplement 2.** Cross-correlation, coherence, and signal-to-noise comparisons for fiber photometry signals.

**Figure supplement 3.** Sustained locomotion-evoked $Ca^{2+}$ signals in nNOS neurons during long duration locomotion events.

nNOS: $0.43 \pm 0.11$, LME difference from CaMKII: $p<0.02$). We also asked if the activity of nNOS neurons were sustained during locomotion, as there is a sustained dilation of arteries during locomotion (*Huo et al., 2015b*; *Gao and Drew, 2016*), and it has been suggested that the response of nNOS neurons is transient (*Iadecola, 2017*). We found that the responses of nNOS neurons were sustained throughout the locomotion bouts (*Figure 7—figure supplement 3* nNOS population GCaMP intensity change in z-units, 10 s: $1.90 \pm 0.46$, LME: $p<0.01$, 15 s: $1.84 \pm 0.52$, LME: $p<0.97$, 30 s: $1.56 \pm 0.40$, LME: $p<0.72$).

When we compared the change in the TRITC fluorescence (a measure of blood volume change) evoked by locomotion in GCaMP-nNOS mice to those in mice expressing GCaMP in pyramidal neurons we found a significantly smaller responses than in mice with calcium indicators in pyramidal neurons (*Figure 7D* Population TRITC intensity change (z-units), hSyn: $0.42 \pm 0.16$, LME difference from CaMKII: $p<0.08$, CaMKII: $0.75 \pm 0.38$, nNOS: $0.46 \pm 0.18$, LME difference from CaMKII: $p<0.04$). Pan-neuronal expression of GCaMP also decreased the CBV change, though not significantly. We do not know the origin of the difference between hemodynamic responses of nNOS and CaMKII-GCaMP mice, but one plausible cause could be that smaller locomotion-evoked response might be caused by the expression of a calcium indicator in the nNOS neurons. Calcium indicators will buffer calcium levels (*Higley and Sabatini, 2008*; *McMahon and Jackson, 2018*), reducing the amplitude of activity-dependent calcium transients by competing with endogenous calcium-binding proteins. This calcium buffering could reduce the production of NO in cells expressing the indicator because the activity of nNOS is calcium-dependent (*Garthwaite, 2008*).

There are several caveats to these experiments. The fluorescence signals from nNOS neurons, whether measured as absolute fluorescence or in the absolute changes in fluorescence, was substantially smaller than the signals from pan-neuronally expressed GCaMP or GCaMP expressed pyramidal neuron localized GCaMP signals. This is a consequence of the much lower number of GCaMP-nNOS neurons than the with the pan-neuronal or pyramidal neuron specific promoters (*Figure 7A–C*). This large difference in the absolute signals for these different conditions means that it would be inappropriate to draw any strong conclusions in differences in the calcium signal amplitudes or in correlation coefficients between blood volumes and neural activity. Calcium indicators are qualitative indicators of neural activity, as simultaneous calcium imaging and electrophysiology has found that the correlation between calcium signals and spiking activity is moderate (*Harris et al., 2016*; *Theis et al., 2016*; *Pachitariu et al., 2018*), and there are likely cell-type specific differences in calcium levels, indicator expression, and coupling of calcium changes to spiking that makes it untenable to make quantitative comparisons across calcium measurements of activity across different cell types. Different neuronal subtypes are known to have differing calcium buffering (*Lee et al., 2000*; *Egger and Stroh, 2009*; *Matthews et al., 2013*) and nitric oxide synthase expressing neurons show low levels of endogenous calcium binding proteins (*Dun et al., 1994*). These results show that nNOS neurons, like pyramidal neurons and other interneurons (*Zhang et al., 2019*), increase their activity during locomotion.

## Relationship between neural activity and arterial dilation

One of the key utilities of hemodynamic imaging is that it allows us to make inferences about neural activity. As we have measured both neural activity and arterial diameter for multiple different perturbations, we sought to understand the relationship between them for different manipulations in pial arteries (*Figure 8—figure supplement 1*). We plotted the arterial diameter and gamma band power

changes for each of the manipulations from the vehicle basal condition in *Figure 8*, with the basal and locomotion conditions connected by a line. A horizontal (left-right) shift means that neural activity is changed without a corresponding change in arterial diameter, while any vertical shift of the points means that there is change in arterial diameter without a corresponding change in neural activity. If there was a perfect, one-to-one ratio between gamma band power and arterial diameter, all of these points would lie along a line or a curve. If this is the case, by knowing the change in arterial diameter, one could infer the change in neural activity accurately from knowing the changes in arterial diameter. At the other extreme, if there were no relationship between neural activity and diameter, the points would be a shapeless blob. In this situation, it is not possible to infer neural activity from a change in arterial dilation.

Within all the manipulations, we found that the increase in neural activity accompanying locomotion was accompanied by an arterial dilation. However, across conditions there was only a weak relationship between neural activity and arterial diameter. For example, the gamma band power was

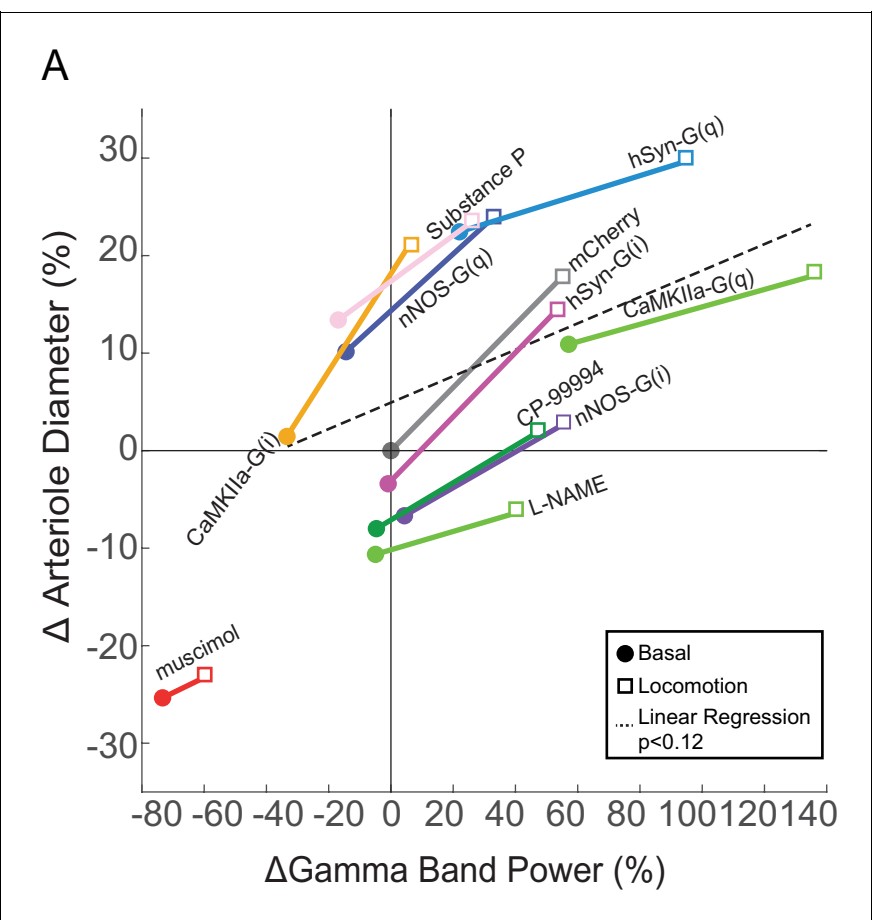

**Figure 8.** Relationship between neural activity and arterial diameter. (**A**) Summary showing the gamma-band power vs. arteriole diameter, normalized to the relevant vehicle control for each condition. Lines connect mean basal and locomotion neural/arterial responses for each condition. Note the large differences in arterial diameter for similar levels of neural activity (e.g. nNOS-G(q) vs. nNOS-G(i)). A linear regression of all the points in the physiologically-relevant range (excluding muscimol), was not significant (p<0.12). For chemogenetic manipulations, neural activity and basal arterial dilation have been shifted so that the reporter virus expressing animals (mCherry) are centered at the origin.

The online version of this article includes the following figure supplement(s) for figure 8:

**Figure supplement 1.** Examples of chemogenetic and pharmacological effects on arteriole diameters.
**Figure supplement 2.** Relationship between arterial diameter and LFP power at lower frequencies.
**Figure supplement 3.** The effects of chemogenetic and pharmacological infusions on evoked arterial diameter changes normalized to the baseline within condition.

similar for both CaMKII-G(i) mice during locomotion and stationary nNOS-G(i) mice, but there were very large (~30%) differences in the arterial diameter. As another example, the change in arterial diameter was the same for stationary CaMKII-G(i) mice as for nNOS-G(i) during locomotion, even though there was an approximately threefold difference in neural activity A linear regression between change in neural activity and change in diameter (excluding the outlier muscimol) reveals no significant relationship between the two (p<0.12). We also looked at the relationship between power in other bands in the LFP (1–10 Hz and 10–40 Hz) and there was either a very weak relationship (for 10–40 Hz, p<0.04, $R^2$ = 0.04) or no significant relationship (1–10 Hz, p<0.05) (*Figure 8—figure supplement 2*). The lack of a strong relationship between arteriole diameter and LFP, which reports the summed activity of all neurons, though with large pyramidal neurons giving a disproportionally large contribution (*Lindén et al., 2011*; *Einevoll et al., 2013*; *Pesaran et al., 2018*), implies that a small set of nNOS expressing neurons can influence arterial diameter without influencing the LFP and vice versa.

## Discussion

Here, we showed that neural activity controls not only evoked arterial dilations, but also basal arterial diameter. Increases or decreases in overall neural excitability drove corresponding basal dilations or constrictions of arterioles. Surprisingly, modulation of the activity of pyramidal neurons had only small effects on basal and evoked arterial dilations. Neurons that express nNOS played an important role in controlling the basal dilation, as chemogenetic inhibition of nNOS positive neurons or NOS inhibition produced a change in basal diameter that was about half as big as seen when silencing activity with muscimol. Although nNOS positive neurons play an important role in controlling arterial diameter, changing their activity makes little contribution to the LFP. These results extend previous work showing that optogenetic stimulation of all interneurons (*Anenberg et al., 2015*; *Uhlirova et al., 2016b*), and nNOS positive interneurons in particular (*Krawchuk et al., 2020*; *Lee et al., 2020*) drive robust vasodilation, with optogenetic stimulation of pyramidal neurons being able to drive only small vasodilations (*Vazquez et al., 2018*). The small effect of manipulations on pyramidal cell activity on basal arterial diameter suggests that nNOS-positive neurons either do not receive strong local pyramidal neuron input, as large changes in pyramidal neuron firing had only small effects on basal and evoked dilations, or these inputs undergo short-term synaptic depression (*Abbott et al., 1997*). When synaptic depression is present at a synapse, it will eliminate the effects of changes in background firing rate (such as those caused by DREADDs) and only transmit the percentage changes in rate. Under normal circumstances in the somatosensory cortex of the awake behaving mouse, the activity of these nNOS-positive neurons is likely correlated to the activity of nearby neurons, as with other neurons, nNOS neurons receive cholinergic modulatory inputs (*Williams et al., 2019*; *Williams et al., 2018*), and cholinergic input increases during locomotion (*Harrison et al., 2016*). Because the activity of nNOS-expressing neurons will be correlated with other neurons, this explains the observed correlations between neural activity and vasodilation. Though increases in blood flow are usually correlated with neural activity, this is not always the case. Hemodynamic signals can occur without corresponding detectable increases in neural activity (*Sirotin and Das, 2009*; *Cardoso et al., 2012*; *Cardoso et al., 2019*), and neural activity increases are not always followed by corresponding increases in blood flow (*Huo et al., 2014*; *Zhang et al., 2019*). In the striatum, increases in neural activity can be accompanied by vasoconstriction and decreases in blood flow (*Shih et al., 2009b*; *Shih et al., 2011*). There are other known mismatches between neural and hemodynamic signals in the visual cortex (*Maier et al., 2008*; *Boynton, 2011*). Our data suggest a mechanism underlying these cases of neurovascular decoupling. It is possible that these mismatches are found when the activity of the vasodilatory nNOS neurons diverges greatly from the rest of the neural population.

These results also show that it is critical to perform quantitative, within-vessel measurements or other calibrations when testing the role of specific signaling pathways on evoked vasodilation. Here, we were able to get absolute measures of arterial diameters for the same vessels under both control and the manipulation conditions. However, hemodynamic signals are often measured ratiometrically relative to the baseline in the given condition, and changes in baseline vessel diameter can confound these ratiometric signals (*Cohen et al., 2002*; *Figure 8—figure supplement 3*). Whenever responses in two different groups of subjects are compared (e.g. transgenic vs. wildtype), the potential for

changes in the baseline are a confound. However, any changes in the baseline in population can be detected by dilating the vessels with isoflurane (*Cudmore et al., 2017*) or $CO_2$ (*Hutchinson et al., 2006*), or testing the basal tone of the vessel in vitro (*Adebiyi et al., 2007*). These tests will reveal if the baseline changes, as the apparent maximal dilation/relaxation will be reduced. Our results show the critical importance of these controls when trying to establish the mechanisms and cellular underpinnings of neurovascular coupling.

There are several caveats to our work. First, the net effect of our chemogenetic manipulations on neural activity depend on the cellular machinery and network interactions in the local neural circuits. Because chemogenetic manipulations alter excitability via endogenous G-protein-coupled ion channels, there will be some variability in their effects between cell-types, and between G(i) and G(q) DREADDs in the same cells, as the amplitude of the excitability modulation depends on the compliment of ion channels present. Our viruses did not infect all nNOS neurons, and were not completely specific. Expression of chemogenetic proteins may also affect basal activity levels in neurons (*Saloman et al., 2016*), which is why whenever possible we have tried to use both pharmacological and chemogenetic manipulations (e.g. G(i)-DREADDs in nNOS neurons and NOS inhibitors) to generate converging lines of evidence. Additionally, increasing and decreasing the activity of a single neuron type does not always have symmetric effects on network activity due to interactions within the cortical circuit (*Phillips and Hasenstaub, 2016*). Similarly, pharmacological effects may not be symmetrical due to levels of baseline activity or modulatory tone. For example, Substance P receptors may have a low level of basal activation in the cortex of awake mice, so blocking them may only have a small effect on the activity of type 1 nNOS neurons in the cortex. In vitro studies have shown that activation of Substance P receptors drives large increases in activity (*Endo et al., 2016*; *Williams et al., 2019*; *Williams et al., 2018*), potentially accounting for the asymmetry of vascular responses to the application of blockers and activators of these receptors. Simulations have shown that the location of NO production relative to the arteriole has a large impact on its efficacy (*Haselden et al., 2020*), so the location of the infected neurons relative to the arterioles may cause some variability. We performed our experiments in awake mice, as anesthetics cause large decreases in brain metabolism, cardiovascular output, and ongoing neural activity (*Gao et al., 2017*), but there could be some variability associated with the mice being awake. However, the locomotion-evoked hemodynamic responses were not due to systemic changes in cardiovascular output, as the locomotion-evoked vasodilation is spatially localized to the FL/HL representation of the somatosensory cortex (*Huo et al., 2014*; *Huo et al., 2015b*; *Huo et al., 2015a*; *Zhang et al., 2019*), not affected by drugs that alter heart rate and blood pressure (*Huo et al., 2015a*), and blocked by local neural silencing (*Figure 1E*). Although we did not monitor heart rates here, head-fixed mice show heart rates comparable to freely moving mice (*Gehrmann et al., 2000*), and fluctuations in heart rates in head-fixed mice do not explain spontaneous and evoked changes in blood flow or vasodilation (*Drew et al., 2011*; *Winder et al., 2017*). None of the manipulations of neural activity used here produced significant changes in the amount of locomotion, so their effects cannot be attributed to changes in behavior. Because many of the manipulations of neural activity (i.e. CaMKIIa-G(i) DREADDs) resulted in minimal changes of arterial diameter across conditions, systemic fluctuations were unlikely to play a substantial role in our preparation. Interestingly, none of our manipulations substantially increased the sensory-evoked dilations above those in control conditions, although some drove large increases in neural activity and/or basal arterial diameter. It seems the dilations that occur during locomotion (~20% above the basal diameter) are near saturation for cerebral arteries under normal circumstances, as increasing the activity of all neurons or nNOS positive neurons were unable to raise the evoked dilation substantially above this level.

Our results suggest a model where approximately half of the dynamic range in the basal and evoked blood arterial diameter is controlled by a small group of neurons and the rest is controlled by other neurons and astrocytes. Any damage to or dysfunction of nNOS neurons could result in decreased basal blood flow, regardless of the metabolic need. Importantly, changes in activity patterns of these neurons will alter basal diameter of arteries, which is known to affect the amplitude of fMRI signals (*Cohen et al., 2002*). Simulations have shown that cerebral arteries account for approximately half of the resistance in the cerebral vasculature (*Schmid et al., 2017*; *Rungta et al., 2018*), so arterial diameter will play an important role in setting basal and evoked blood flow levels. There is a decline in the number of interneurons, particularly nNOS-expressing interneurons, with aging (*Miettinen et al., 1993*; *Necchi et al., 2002*), and the removal of their tonic vasodilatory signal could

contribute to decreases in cerebral blood flow that are thought to lead to dementia (*Wolters et al., 2017*). Indeed, alterations of nNOS expression have been implicated in Alzheimer's disease (*Han et al., 2019b*) and chronic stress (*Han et al., 2019a*). Protecting these neurons from insults and damage could be a promising strategy for preventing neurovascular disorders.

# Materials and methods

## Key resources table

| Reagent type (species) or resource | Designation | Source or reference | Identifiers | Additional information |
|---|---|---|---|---|
| Strain, strain background (*Mus musculus*) | C57BL/6J | Jackson Labs | #000664 | |
| Strain, strain background (*Mus musculus*) | B6.129-Nos1$^{tm1(cre)Mgmj}$/J | Jackson Labs | #017526 | |
| Strain, strain background (*Mus musculus*) | FVB-Tg(*Aldh1l1*-cre/ERT2)1Khakh/J | Jackson Labs | #023748 | |
| Strain, strain background (*Mus musculus*) | C57BL/6-Tg(CAG-EGFP)131Osb/LeySopJ | Jackson Labs | #006567 | |
| Genetic reagent (Adeno-associated virus) | AAV5-CMV-Turbo RFP-WPRE-rBG | UNC vector core | | |
| Genetic reagent (Adeno-associated virus) | AAV8-hSYN-HA-hM3D (Gq)-mCherry | Addgene | #50474-AAV8 | |
| Genetic reagent (Adeno-associated virus) | AAV5-hSYN-HA-hM4D (Gi)- mCherry | UNC Vector Core | | |
| Genetic reagent (Adeno-associated virus) | AAV5-CaMKIIa-hM3D (Gq)- mCherry | Addgene | #50476-AAV5 | |
| Genetic reagent (Adeno-associated virus) | AAV5-CaMKIIa-hM4D (Gi)- mCherry | UNC Vector Core | | |
| Genetic reagent (Adeno-associated virus) | AAV8-DIO-hM3D (Gq)-mCherry | Addgene | #44361-AAV8 | |
| Genetic reagent (Adeno-associated virus) | AAV8-DIO-hM4D (Gi)-mCherry | Addgene | #44362-AAV8 | |
| Genetic reagent (Adeno-associated virus) | AAV9-CaMKII-GCaMP6s | Addgene | #107790-AAV9 | |
| Genetic reagent (Adeno-associated virus) | AAV8-syn-GCaMP-WPRE | Vigene | AAV8-syn-GCaMP-WPRE | |
| Genetic reagent (Adeno-associated virus) | AAV9-CaMKII-GCaMP6s | Addgene | #107790-AAV9 Viagene | |
| Antibody | Mouse monoclonal GAD-65 Antibody (A-3) | Santa Cruz Biotechnology | sc-377145 | (1:200) |
| Antibody | Mouse monoclonal CaMKII Antibody (G-1) | Santa Cruz Biotechnology | sc-5306 | (1:200) |
| Antibody | Rabbit polyclonal to GFAP | Abcam | ab7260 | (1:200) |
| Antibody | Alexa Fluor 488 Goat Anti-Mouse IgG H and L | Abcam | ab150113 | (1:500) |
| Antibody | Goat Anti-Rabbit IgG H and L | Abcam | ab6702 | (1:500) |
| Antibody | Goat polyclonal nNOS | Abcam | ab1376 | (1:200) |
| Chemical compound, drug | muscimol | Sigma-Aldrich | M1523 | 10 mM |
| Chemical compound, drug | L-NAME | Sigma-Aldrich | N5751 | 1 mM |
| Chemical compound, drug | Substance P | Tocris | 1156 | 1 mM |
| Chemical compound, drug | CP-99994 | Tocris | 3417 | 8 mM |
| Chemical compound, drug | ML-133 | Sigma-Aldrich | SML0190 | 10 μM |
| Chemical compound, drug | DMSO | Sigma-Aldrich | D8418 | 1% |

*Continued on next page*

*Continued*

| Reagent type (species) or resource | Designation | Source or reference | Identifiers | Additional information |
|---|---|---|---|---|
| Chemical compound, drug | $BaCl_2$ | EMD Millipore | B1493816 | 100 µM |
| Chemical compound, drug | CNO | Sigma-Aldrich | C0832 | 2.5 mg/kg |
| Chemical compound, drug | FITC-dextran 70 kDa | Sigma-Aldrich | 46945 | 50 µL at 5% |
| Chemical compound, drug | TRITC dextran 70kD | Sigma-Aldrich | T1162 | 50 µL at 5% |
| Software, algorithm | Pial vessel dimeter measurements | https://github.com/DrewLab/Surface-Vessel-FWHM-Diameter | | |
| Software, algorithm | Penetrating vessel dimeter measurements | https://github.com/DrewLab/Thresholding_in_Radon_Space | | |
| Software, algorithm | Red blood cell velocity measurements | https://github.com/DrewLab/MCS_Linescan | | |
| Software, algorithm | 2 P microscope control software | MCS, Sutter Instruments, Novato CA | | |
| Software, algorithm | MATLAB | Mathworks, Natick MA | | |
| Software, algorithm | Doric Neuroscience Studio | Doric Lenses, Quebec, Quebec Canada | | |

## Animal procedures

This study was performed in strict accordance with the recommendations in the Guide for the Care and Use of Laboratory Animals of the National Institutes of Health. All procedures were performed in accordance with protocols approved by the Institutional Animal Care and Use Committee (IACUC) of Pennsylvania State University (protocol # 201042827). All surgeries were performed under isoflurane anesthesia and every effort was made to minimize suffering. Data were acquired from 209 mice (107 mice for 2PLSM imaging, 61 mice for electrophysiology, 21 mice for fiber photometry, and 20 mice for immunohistochemistry - see *Tables 1* and *2*) C57BL/6J mice, cre-nNOS mice (B6.129-*Nos1*^tm1(cre)Mgmj/J), C57BL/6-Tg(CAG-EGFP), and *Aldh1L1*-cre-mice (B6;FVB-Tg(*Aldh1l1*-cre/ERT2) 1Khakh/J Jackson Laboratory) between 3 and 8 months of age were used. Some mice received infusions of multiple drugs. Mice were given food and water ad libitum and maintained on 12 hr light/dark cycles in isolated cages during the period of experiments. All in vivo experiments were performed between 8:00 and 12:00 ZT. Sample size was chosen to be consistent with previous studies in the literature (*Zhang et al., 2019*; *Aydin et al., 2020*).

### Virus injections

All viruses were obtained from UNC Vector Core, Addgene, or Vigene (see *Table 1*). For viral injections, mice were anaesthetized with isoflurane (5% induction, 2% maintenance) and placed in a stereotaxic rig. A small craniotomy was made over the FL/HL representation of the somatosensory cortex (0.75 mm caudal, 2.5 mm lateral from bregma). Using a pulled glass micropipette (30–50 µm tip diameter) and infusion pump (Harvard Apparatus, Holliston, MA), we microinjected 500 nanoliters (at a rate 100 nl/min) of the adeno-associated viral (AAV) vectors (see *Table 1*) ~300 µm beneath the cortical surface. After the viral injections were completed, the glass micropipette was kept in place for 10 min before withdrawal. The skin was sutured, and mice were returned to their home cage. Four weeks after AAV injections, we performed surgeries for electrode or window implantation, or perfusion for immunohistochemistry.

### Window, electrode, and cannula implantation procedures

Mice were anesthetized with isoflurane and the scalp was resected. A custom-made titanium metal bar (https://github.com/DrewLab/Mouse-Head-Fixation) was fixed to the skull with cyanoacrylate glue (Vibra-Tite, 32402) and dental cement just posterior to the lambda cranial suture. A 4–8 mm² area polished and reinforced thinned-skull window was created over the forepaw/hindpaw representation of somatosensory cortex on the right hemisphere, as described previously (*Drew et al., 2010b*; *Gao and Drew, 2016*; *Winder et al., 2017*). The thinned area was polished with size 3F grit (Covington Engineering, Step Three 3 F-400) and reinforced with a fitted #0 glass coverslip

**Table 1.** AAV-injected mice.

| AAV-injected mice | # mice (vessels) for 2-photon imaging | # mice for chronic electrophysiology | Immunohistochemistry | Fiber photometry |
|---|---|---|---|---|
| C57BL/6J: AAV5-CMV-TurboRFP-WPRE-rBG (UNC Vector Core) | 12 mice (56 vessels) | Four mice | Three mice | |
| C57BL/6J: AAV8-hSYN-HA-hM3D(Gq)-mCherry (Addgene #50474-AAV8) | Seven mice (37 vessels) | Three mice | Three mice | |
| C57BL/6J: AAV5-hSYN-HA-hM4D(Gi)- mCherry (UNC Vector Core) | Seven mice (45 vessels) | Three mice | Two mice | |
| C57BL/6J: AAV5-CaMKIIa-hM3D(Gq)- mCherry (Addgene # 50476-AAV5) | Six mice (41 vessels) | Four mice | Two mice | |
| C57BL/6J: AAV5-CaMKIIa-hM4D(Gi)- mCherry (UNC Vector Core) | Seven mice (38 vessels) | Five mice | Two mice | |
| B6.129-Nos1$^{tm1(cre)Mgmj}$/J: AAV8-DIO-hM3D(Gq)-mCherry in nNOS-cre mice (Addgene #44361-AAV8) | Eight mice (52 vessels) | Four mice | Two mice | |
| B6.129-Nos1$^{tm1(cre)Mgmj}$/J: AAV8-DIO-hM4D(Gi)-mCherry in nNOS-cre mice (Addgene #44362-AAV8) | Nine mice (49 vessels) | Five mice | Two mice | |
| B6;FVB-Tg(Aldh1l1-cre/ERT2)1Khakh/J: AAV8-DIO-hM4D(Gi)-mCherry in astrocyte-cre mice (Addgene #44362-AAV8) | Five mice (24 vessels) | Three mice | Two mice | |
| B6;FVB-Tg(Aldh1l1-cre/ERT2)1Khakh/J: AAV8-DIO-hM4D(Gi)-mCherry in astrocyte -cre mice (Addgene #44362-AAV8) | Seven mice (31 vessels) | Four mice | Two mice | |
| C57BL/6J: AAV9-CaMKII-GCaMP6s (Addgene) | | | | Five mice |
| C57BL/6J: AAV8-syn-GCaMP-WPRE (Vigene) | | | | Four mice |
| B6.129-Nos1$^{tm1(cre)Mgmj}$/J: AAV9-CaMKII-GCaMP6s (Addgene) | | | | Nine mice |
| C57BL/6-Tg(CAG-EGFP)131Osb/LeySopJ (Jackson Labs) | | | | Three mice |

(Electrode Microscopy Sciences, #72198). Self-tapping screws (#000, 3/32', JI-Morris, Southbridge, MA) were placed into the contralateral parietal and ipsilateral frontal bone to stabilize the skull. The titanium bar and screws were secured with black dental acrylic resin. The animals were allowed 2–3 days to recover before habituation.

For electrode or cannula implantation, small (<0.5 mm diameter) craniotomies were made to insert the stereotrode or cannula (Plastics One, C315DCS, C315GS-4) into the upper layers of cortex near the FL/HL representation of cortex. The cannula was attached to the skull and headbar with cyanoacrylate glue and dental acrylic. Implanted cannulas do not alter the hemodynamic response (Winder et al., 2017). The position of the cannula within the window was located relative to the large surface vessels, which are clearly visible on wide-field images taken under a surgical

**Table 2.** Infused mice.

| Infused mice | # mice (vessels) for two-photon imaging | # mice for chronic Electrophysiology |
|---|---|---|
| C57BL/6J: Muscimol (10 mM) | Nine mice (26 vessels) | Three mice |
| C57BL/6J: L-NAME (1 mM) | Six mice (36 vessels) | Five mice |
| C57BL/6J: Substance P (1 μM) | Seven mice (30 vessels) | Four mice |
| C57BL/6J: CP-99994 (8 mM) | Seven mice (28 vessels) | Four mice |
| C57BL/6J: ML-133 (10 μM) | Five mice (36 vessels) | Four mice |
| C57BL/6J: BaCl$_2$ (100 μM) | Five mice (24 vessels) | Three mice |

microscope. The position of the vessels imaged under the two-photon microscope relative to the cannula tip using the major surface vessels as landmarks.

## Habituation

Mice were habituated to head-fixation on the spherical treadmill (60 mm diameter) over three days before imaging. Mice were habituated for 15 min of acclimation on the first day. On successive days, the time was increased to up to 2 hr for 3 to 4 days before imaging/electrophysiology/fiber photometry. During habituation, mice were monitored for any signs of distress. 2PLSM data were taken over four imaging sessions, with two sessions for each condition.

## Intraperitoneal injections and cortical infusions

CNO (2.5 mg/kg in 2% DMSO in saline) or 2%DMSO in saline were injected intraperitoneal 20 min before imaging began. Pilot electrophysiological experiments showed that the effects of DREADDs on neural activity onset within 20 min of injection. Injections of CNO and DMSO were counterbalanced.

For the intracortical infusions, mice were first head-fixed on the spherical treadmill. The dummy cannula was then slowly removed and replaced with an infusion cannula (Plastics One, C315IS-4). The interface between the infusion cannula and the guide cannula was sealed with Kwik-Cast (World Precision Instruments). All infusions were done at a rate of 25 nL/min for a total volume of 500 nL. Animals were kept in place for 20 min after the end of the infusion and then moved to the imaging apparatus. The infusion cannulas were kept in place for the duration of the imaging session. Pharmacological treatments and aCSF were each infused in a counterbalanced order.

## Physiological measurements

### Two-photon microscopy

Prior to imaging, mice (n = 107 mice) were briefly anesthetized with isoflurane, and retro-orbitally injected with 50 µL 5% (weight/volume) fluorescein conjugated dextran (FITC-dextran 70 kDa; Sigma-Aldrich) (*Drew et al., 2011*; *Shih et al., 2012b*) then head-fixed upon a spherical treadmill. The treadmill was coated with anti-slip tape and attached to an optical rotary encoder (US Digital, E7PD-720–118) to monitor rotational of the treadmill. Imaging was done on a Sutter Movable Objective Microscope with a $16 \times 0.8$ NA objective (Nikon). A MaiTai HP laser (Spectra-Physics) tuned to 800 nm was used to excite the FITC. The power exiting the objective was 10–15 mW for imaging surface arteries. For vessel diameter measurements, movies of individual arteries were taken at a nominal frame rate of 8 Hz for 5 min. For RBC velocity measurements, line-scans were made along the long axis of the capillary lumen (Figure S4A) (*Drew et al., 2010a*). The same vessel segments were imaged after both the treatment (intraperitoneal CNO injection or local drug infusion) and vehicle controls (intraperitoneal vehicle injection or intracranial vehicle infusion). Treatments and vehicle condition were imaged in a counterbalanced order on separate days. Imaged arteries were chosen in the first session of imaging. Arteries were visually identified by their more rapid blood flow, rapid temporal dynamics of their response to locomotion, and vasomotion (*Drew et al., 2010b*; *Drew et al., 2011*; *Winder et al., 2017*). Bright field and fluorescent images of the PoRTs window were taken before imaging on the two-photon for each mouse. For imaging experiments with DREADDs manipulations, vessels were selected for imaging if they were within the region expressing fluorescent reporter protein (*Figure 2A*). For infusion imaging experiments, selected vessels were less than 1.5 mm from the cannula. For repeated imaging of the same vessel, three-dimensional image stacks of the regions were made, and the position of nearby vessels were used to return to the same imaging plane on subsequent imaging sessions. Vessels within the targeted region were chosen on the first (and third for DREADD experiments) imaging session randomly, and the analysis was done with automated software, effectively blinding the experimenter. Only vessels anatomically identified to be in the FL/HL representation were used in locomotion-triggered averages.

### Image processing and data analysis

Data analysis was performed in MATLAB (MathWorks). 2PLSM images were aligned in the x–y plane using a rigid-body registration algorithm (*Drew et al., 2011*). Each movie was visually reviewed to ensure that z-axis motion during locomotion was minimal. For surface arteries, the intensity of a

short segment (1–3 µm in length) was averaged along the long axis of the vessel, and the diameter was calculated from the full-width at half-maximum (https://github.com/DrewLab/Surface-Vessel-FWHM-Diameter, *Drew et al., 2011*). For penetrating arteries, the Thresholding in Radon Space (TiRS) method was used as it effectively performs a full-width at half maximum measurement along every angle of the penetrating artery (https://github.com/DrewLab/Thresholding_in_Radon_Space, *Gao and Drew, 2014*). For diameters of penetrating vessels, using the TiRS method is important as penetrating vessels are not circular and the cross sectional shape will change during dilation, making single-axis diameter measurements inaccurate (*Greensmith and Duling, 1984*; *Steelman et al., 2010*; *Gao and Drew, 2014*). A square region of interest (ROI) enclosing the penetrating arteriole of interest was manually drawn. The images were transformed into Radon space, thresholded, and then transformed back to image space, where the vessel cross-sectional area was quantified after a second thresholding. In order to facilitate comparison with pial vessels, diameters (D) of penetrating vessels were taken to be: $D = 2\sqrt{A/\pi}$, where A was the cross-sectional area calculated using the TiRS method. Frames in which the temporal derivative of diameter changed by >16 µm/s were tagged as motion artifacts and replaced with the linear interpolation between the proceeding and subsequent points. The diameter was filtered with a five-point median filter (MATLAB function: medfilt1; filter order = 5). Red blood cell velocity in capillaries was calculated using the Radon transform (*Drew et al., 2010a*) (https://github.com/DrewLab/MCS_Linescan). Example images of individual vessels were taken from movies that were 3D median filtered in ImageJ (ImageJ: 3D Median, radius = 2) (the third dimention here being time), and then the max intensity projection was taken for 40 frames with or without locomotion.

To detect locomotion events, the velocity signal from the rotational encoder on the spherical treadmill was low-pass filtered (10 Hz, fifth-order Butterworth), and then the absolute value of the acceleration was binarized with a $10^{-5}$ cm/s$^2$ threshold (*Huo et al., 2014*; *Huo et al., 2015b*). Stationary periods were defined as periods when the mouse was still, with at least a 2 s buffer after the end of any preceding locomotion event and a 1 s buffer before the start of the next locomotion event. Onset time was calculated by finding the time intercept of a line fit through the points in the locomotion-triggered average that were 20% and 80% of the peak dilation (*Tian et al., 2010*). The change in basal diameter in each treatment condition was normalized by the basal diameter of the same vessel in the vehicle control condition unless otherwise specified.

For the vessels that were histologically identified as being within the FL/HL representation, locomotion-triggered averages (LTA) were calculated by taking the average of any locomotion event >5 s in duration with at least 2 s of no locomotion before the event. These averages were normalized by the average basal diameter of the same vessel under vehicle control conditions.

## Electrophysiology

Neural activity was recorded as differential potentials between the two leads of Teflon-insulated tungsten micro-wires (A-M Systems, #795500) (*Huo et al., 2014*; *Winder et al., 2017*). Differential recordings between two closely spaced electrodes avoid the volume conduction of remote signals (*Parabucki and Lampl, 2017*). Stereotrode micro-wires, with an inter-electrode spacing of ~100 µm, were threaded through polyimide tubing (A-M Systems, #822200). Electrode impedances were between 70 and 120 kΩ at 1 kHz. The electrodes were implanted in the upper layers of cortex (~400 µm depth). The acquired signals were amplified (World Precision Instruments, DAM80), band-pass filtered between 0.1 and 10 kHz (Brownlee Precision, Model 440) during acquisition, then digitized at 20 kHz.

Power was calculated after digitally band-pass filtering the raw neural signal (MATLAB function: butter, filtfilt; filter order = 4). The local field potential (LFP) was calculated by digitally band-pass filtering the raw neural signal between 10–100 Hz (MATLAB function: butter, filtfilt; filter order = 4) and using a notch filter to remove 60 Hz noise (MATLAB function: iirnotch). The basal power was computed by calculating the power during periods of no locomotion. The locomotion power was computed by calculating the power during periods of locomotion. Gamma-band power was quantified by averaging the 40–100 Hz band of the LFP after using a notch filter to remove 60 Hz noise.

## Fiber photometry

Mice were injected with 0.5 µL of AAV with a GCaMP6s construct (AAV9-CaMKII-GCaMP6s-WPRE-Sv40 from Addgene, or AAV8-syn-GCaMP-WPRE or AAV8-FLEX-syn-GCaMP-WPRE from Vigene). After a month of recovery, mice underwent surgery to implant a titanium head bar and a fiber optic canula (1.5 mm length, 200 µm diameter, 0.48NA, Doric Lenses, Quebec, Canada) in the FL/HL representation. Two days later, the mice began habituation to being headfixed on the spherical treadmill of the same design used for two-photon imaging. Four days of habituation took place before any recordings were made. For fiber photometry measurements, we used a Doric Lenses system with LEDs illuminating at 405, 465, and 560 nm, and collected fluorescence between 500 and 540 nm (GFP or GCaMP) and 580–680 nm (TRITC) at 12 kHz which was then down-sampled to 1.2 kHz. Prior to measurements, mice were briefly anesthetized with isoflurane and 50 µL of 5% TRITC dextran (70kD, Sigma-Aldrich) was injected into the retroorbital sinus. Data was acquired starting 20 min after the injection. Data from each mouse was taken over three sessions (~2 hr each) with the Doric Neuroscience studio software.

## Fluorescence correction and Z-scoring

All data processing was done in MATLAB. Locomotion data was zero-lag low-pass filtered below 10 Hz and then binarized into locomotion or not locomoting, using a threshold set based on the absolute value of the first derivative of the signal. To correct for excretion of TRITC and photobleaching of GCaMP signals, raw fluorescence signals were zero-lag low pass filtered below 0.1 Hz and periods of locomotion and the following fifteen seconds were excluded from the data set to avoid contamination of metabolic decay with behavior-evoked blood volume changes. This data was then fit with the sum of two exponential decays which was subsequently subtracted from the raw broadband signal. The corrected signals were then zero-lag low pass filtered below 1 Hz. To correct for variable signal intensities, low pass filtered signals were then normalized to have a minimum of 0 and maxima of 1. Because increases in local hemoglobin concentration attenuate GCaMP signals, correction of GCaMP signals with respect to the acquired blood volume signal was performed. Three animals expressing GFP ubiquitously (*Okabe et al., 1997*) were used to determine the relationship between changes in blood volume and the fluorescence signal. The relationship between the green fluorescence and CBV was modeled using a linear function and the slopes of the three animals were averaged to determine the mean attenuation coefficient. To correct the effect of CBV changes on GCaMP fluorescence, TRITC signal was multiplied by this coefficient and then subtracted from the GCaMP channel, removing any effects of increased hemoglobin on fluorescence. All GCaMP data was then zero-lag bandpass filtered between [0.001 Hz and 1 Hz]. Fluorescence intensity was then converted to z-scores. Locomotion events were defined as periods of locomotion at least 5 s in duration with at least 5 s preceding locomotion onset free of any behavior to establish resting baseline of hemodynamic and GCaMP signals. Locomotion events were separated in to five groups based on their duration for analysis, [5.0–9.99 s], [10–14.99 s], [15–29.99 s], [30.00–44.99 s], and [45.0s+]. A baseline of the average intensity preceding each locomotion event was subtracted from each event, and averages were calculated within each category.

## Histology

At the end of the experiment, mice were deeply anesthetized with 5% isoflurane, and then transcardially perfused with heparinized saline followed by 4% paraformaldehyde. The brain was removed from the skull and sunk in 30% sucrose/4% PFA. For cytochrome oxidase staining, the cortex was then flattened and 60 µm tangential sections were cut on a freezing microtome. The cytochrome oxidase in the sections was stained, and the location of the FL/HL representation in somatosensory cortex was reconstructed relative to the vasculature visible through the cranial window (*Adams et al., 2018*).

For immunohistochemistry labeling, the brains were immersed in a 4% PFA/30% sucrose solution for 1 day. Tissues were sectioned coronally to a 90 µm thickness and then washed in PBS twice. For heat-induced antigen retrieval, the slices were boiled in 10 mM sodium citrate for 10 min. The sections were then incubated with primary antibodies on slides at a 1:200 dilution for two days at 4°C (Mouse monoclonal GAD-65 Antibody (A-3): sc-377145 Santa Cruz Biotechnology, Mouse monoclonal CaMKII Antibody (G-1): sc-5306 Santa Cruz Biotechnology, Rabbit polyclonal to GFAP: ab7260

abcam, Goat polyclonal anti-nNOS: ab1376 Abcam). The brain sections were washed twice and incubated for one hour with secondary antibodies (Abcam Alexa Fluor 488 Goat Anti-Mouse IgG H and L and Goat Anti-Rabbit IgG H and L). The brain sections were washed twice, and coverslipped with DAPI fluoroshield mounting media (Abcam ab104139). The sections were imaged on an Olympus FV10i Confocal. Manual quantification of cell specificity were done in 318 μm x 318 μm x 120 μm volumes. Type 1 nNOS neurons were distinguished from type two neurons by their substantially more intense staining for nNOS (*Perrenoud et al., 2012*).

## Statistical analysis

Statistical analysis was performed using MATLAB (R2018, MathWorks). All summary data were reported as mean ± standard deviation and plotted as mean ± standard error unless otherwise specified. For comparisons of vehicle vs. treatment conditions, where multiple measurements of different vessel diameters were taken from the same mouse, we used the linear mixed effects (LME) model (MATLAB function: fitlme). For each manipulation (e.g. CNO vs. vehicle in DREADD-expressing mice), we fit a linear mixed effects model given by:

$$D_{n,a} = \gamma + \beta_a$$

where $D_{n,a}$ is the diameter differences between the vehicle and manipulation condition, $a$ is the identifier of the animal group, and $n$ is the identifier of the vessel. The $\gamma$ term is the (fixed) effect of the manipulation, and $\beta_a$ is the within animal (random) effect that accounts for the within animal correlations (*Aarts et al., 2014*). To determine if the manipulation caused a significant change in the vessel diameter, we calculated if the $\gamma$ term differed from 0, after a Bonferroni correction for the number of groups of DREADD-expressing mice (7 groups) or the number of infusions (6 groups). p-Values >1 after correction were rounded down to 1. A difference was determined to be statistically significant if p<0.05 after Bonferroni correction. For plotting the relationship of the basal diameter in the control condition versus the manipulation in the figures, we fit a linear mixed effects model given by:

$$D_{n,a} = \gamma + \beta_a + \alpha_n D_n$$

where $D_n$ is the diameter of vessel $n$ in the control condition, and $\alpha_n$ is a vector of (fixed) coefficients.

For comparisons of vehicle vs. treatment conditions of the gamma-band (averaged power from 40 to 100 Hz after removing 60 Hz noise), we used the paired t-test (MATLAB function: ttest).

## Acknowledgements

This work was supported by grants R01NS078168 and R01NS101353 from the NIH to PJD and F31NS105461 to JNN. We thank A Winder, and Q Zhang for help with experiments, and Y Kim and N Zhang for comments on the manuscript.

## Additional information

### Funding

| Funder | Grant reference number | Author |
|---|---|---|
| National Institute of Neurological Disorders and Stroke | R01NS078168 | Patrick J Drew |
| National Institute of Neurological Disorders and Stroke | R01NS101353 | Patrick J Drew |
| National Institute of Neurological Disorders and Stroke | F31NS105461 | Jordan N Norwood |

The funders had no role in study design, data collection and interpretation, or the decision to submit the work for publication.

## Author contributions
Christina T Echagarruga, Conceptualization, Data curation, Software, Formal analysis, Investigation, Visualization, Methodology, Writing - original draft, Writing - review and editing; Kyle W Gheres, Conceptualization, Resources, Data curation, Software, Formal analysis, Validation, Investigation, Visualization, Methodology, Writing - original draft, Writing - review and editing; Jordan N Norwood, Funding acquisition, Investigation, Visualization, Writing - original draft; Patrick J Drew, Conceptualization, Software, Formal analysis, Supervision, Funding acquisition, Validation, Investigation, Visualization, Methodology, Writing - original draft, Project administration, Writing - review and editing

## Author ORCIDs
Kyle W Gheres (iD) https://orcid.org/0000-0001-7568-9023
Jordan N Norwood (iD) http://orcid.org/0000-0001-8093-5938
Patrick J Drew (iD) https://orcid.org/0000-0002-7483-7378

## Ethics
Animal experimentation: This study was performed in strict accordance with the recommendations in the Guide for the Care and Use of Laboratory Animals of the National Institutes of Health. All procedures were performed in accordance with protocols approved by the Institutional Animal Care and Use Committee (IACUC) of Pennsylvania State University (protocol # 201042827). All surgeries were performed under isoflurane anesthesia and every effort was made to minimize suffering.

## Decision letter and Author response
Decision letter https://doi.org/10.7554/eLife.60533.sa1
Author response https://doi.org/10.7554/eLife.60533.sa2

# Additional files

## Supplementary files
• Transparent reporting form

## Data availability
The Matlab code and data to generate the figures have been uploaded to Dryad. The DOI for download is here: https://doi.org/10.5061/dryad.b8gtht79h.

The following dataset was generated:

| Author(s) | Year | Dataset title | Dataset URL | Database and Identifier |
|---|---|---|---|---|
| Echagarruga CT, Gheres K, Norwood JN, Drew PJ | 2020 | Data from: nNOS-expressing interneurons control basal and behaviorally-evoked arterial dilation in somatosensory cortex of mice | https://doi.org/10.5061/dryad.b8gtht79h | Dryad Digital Repository, 10.5061/dryad.b8gtht79h |

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
