## [Decision Letter]

**Acceptance summary:**

Brain activity is tightly coupled to increases in blood flow, a phenomenon that is exploited by neuroimaging techniques based on blood oxygenation to study brain function. This paper elegantly demonstrates that increases in blood flow do not actually reflect overall activity of the region but rather are controlled by a minor subset of specialized inhibitory neurons, providing important new insights into how neuroimaging data are interpreted. This paper is likely to be of interest to a broad range of readers, both researchers who use animal models as well as human brain imaging researchers. The paper will also likely be of interest to those researchers who are interested in the functions of different types of interneurons and those who study activity-dependent changes in blood flow in the brain.

**Decision letter after peer review:**

Thank you for submitting your article "nNOS-expressing interneurons control basal and behaviorally-evoked arterial dilation in somatosensory cortex of mice" for consideration by *eLife*. Your article has been reviewed by three peer reviewers, including Anusha Mishra as the Reviewing Editor and Reviewer #1, and the evaluation has been overseen by Laura Colgin as the Senior Editor. The following individuals involved in review of your submission have agreed to reveal their identity: Clare Howarth (Reviewer #2); Ian Kimbrough (Reviewer #3).

The reviewers have discussed the reviews with one another and the Reviewing Editor has drafted this decision to help you prepare a revised submission.

Summary:

This careful and well-controlled study by Echagarruga et al. combines chemogenetic and pharmacological approaches with 2-photon microscopy in awake, behaving mice to demonstrate that neuronal nitric oxide synthase (nNOS) expressing interneurons, rather than pyramidal excitatory neurons, are the major regulators of basal and evoked changes in arterial diameter in the rodent somatosensory cortex. The work addresses an important gap in our knowledge, addressing the long-standing question of which neural subpopulations regulate arterial diameter to produce functional hyperemia. Other groups have recently published similar findings suggesting a dominant role of inhibitory, particularly nNOS-expressing, interneurons in vasoregulation. This study further elucidates the role of nNOS neurons as drivers arteriole tone, both at baseline and during changes evoked by behavior, and narrows this down specifically to type I nNOS interneuron subtype. This study is well executed, is timely, and has implications for our understanding of functional neuroimaging techniques and diseases associated with impaired vascular regulation (e.g. Alzheimer's disease).

Essential revisions:

All the reviewers agree that this study represents a significant step forward in the field. There are a few concerns, as described in the reviews below, which require revisions prior to acceptance. The essential points for revision are:

1) Defining the effect of each manipulations on pial vs. penetrating arterioles – most figures show pial arteriole data, and some supplementary figures show penetrating arteriole data, but the text does not distinguish between them. It would be useful to weave these together to discuss how the contribution of each arteriole type is integrated into the overall response. In the case that an effect is observed in pial vessels but not in penetrating arterioles (e.g. for nNOS neuron excitability in Figure 4 vs. Figure 1—figure supplement 2N-Q), a discussion of how this could occur would enhance the manuscript.

2) Quantification of the specificity of DREADD expression would strengthen the manuscript. What % of DREADD+ cells were positive for the target neuron population (e.g. DREAD+/nNOS+) and what % were negative (e.g. DREAD+/nNOS-)? It would be good if this could be done for all neuron-specific expression, but is particularly important for the nNOS data. More information on whether nNOS type I and nNOS type II neurons were distinguished and, if so, how, would also be helpful.

3) While we generally agree with the author's justification for normalizing their measurements to the vehicle-treated baseline, the way the data are reported allow too much room for potential confusion. E.g. in the case of muscimol, where locomotion-evoked change in dilation is reported as -21.6% (subsection “Basal arterial diameter and evoked dilation are controlled by local neural activity”, last paragraph), this can easily be misinterpreted as a negative dilation or a constriction, although it is clear from the graph (Figure 1E) that there is a dilation and not a constriction. The data being normalized to the vehicle is perfectly sensible, but the difference in diameter that is normalized should be the difference between the locomotion-evoked to baseline diameter under the same manipulation (e.g., in muscimol for the example cited above), which would avoid negative dilation numbers.

---

## [Author Response]

Essential revisions:All the reviewers agree that this study represents a significant step forward in the field. There are a few concerns, as described in the reviews below, which require revisions prior to acceptance. The essential points for revision are:1) Defining the effect of each manipulations on pial vs. penetrating arterioles – most figures show pial arteriole data, and some supplementary figures show penetrating arteriole data, but the text does not distinguish between them. It would be useful to weave these together to discuss how the contribution of each arteriole type is integrated into the overall response. In the case that an effect is observed in pial vessels but not in penetrating arterioles (e.g. for nNOS neuron excitability in Figure 4 vs. Figure 1—figure supplement 2N-Q), a discussion of how this could occur would enhance the manuscript.

To address the reviewers’ concerns #1 and #3, we have added a new section “Comparison of pial and penetrating arteriole basal and evoked diameter changes”, in which we compare the basal changes in diameter of pial and penetrating arterioles, as well as the change in diameter from the baseline during locomotion. We found that penetrating vessels show changes in basal diameter that are consistent with those in pial vessels, but that the none of the manipulations of neural activity have significant effects on the evoked changes in penetrating arteriole diameter (Figure 6—figure supplement 2). Consistent with previous work (Gao et al., 2015), we found significantly smaller dilations of penetrating vessels during locomotion relative to pial arterioles. The difference between pial and penetrating arterioles could be due to a difference in the electrical contractile properties of the pial vs. penetrating vessels, the location of the vasodilatory signal release (Haselden, Kedarasetti and Drew, 2020), or a mechanical restriction of dilation by the surrounding brain tissue (Fung, Zweifach and Intaglietta, 1966; Liu et al., 2007), all of which we discuss.

2) Quantification of the specificity of DREADD expression would strengthen the manuscript. What % of DREADD+ cells were positive for the target neuron population (e.g. DREAD+/nNOS+) and what % were negative (e.g. DREAD+/nNOS-)? It would be good if this could be done for all neuron-specific expression, but is particularly important for the nNOS data. More information on whether nNOS type I and nNOS type II neurons were distinguished and, if so, how, would also be helpful.

We have added quantification of the DREADD expression in nNOS neurons to the section “nNOS-expressing neurons and NO signaling control arterial diameter independent of overall neural activity” and details of the quantification to the “Histology” section of the Materials and methods. Approximate one third of 1 neuron and Type 2 neurons expressed DREADDs, and about 45% of the DREADD expressing cells were positive for nNOS.

3) While we generally agree with the author's justification for normalizing their measurements to the vehicle-treated baseline, the way the data are reported allow too much room for potential confusion. E.g. in the case of muscimol, where locomotion-evoked change in dilation is reported as -21.6% (subsection “Basal arterial diameter and evoked dilation are controlled by local neural activity”, last paragraph), this can easily be misinterpreted as a negative dilation or a constriction, although it is clear from the graph (Figure 1E) that there is a dilation and not a constriction. The data being normalized to the vehicle is perfectly sensible, but the difference in diameter that is normalized should be the difference between the locomotion-evoked to baseline diameter under the same manipulation (e.g., in muscimol for the example cited above), which would avoid negative dilation numbers.

We agree that reporting the change from the baseline is an important metric, and have added these absolute diameter changes to the section “Comparison of pial and penetrating arteriole basal and absolute diameter changes” and a figure supplement (Figure 6—figure supplement 2) showing this and performing statistical comparisons. We have also added additional text clarifying that the reported numbers are relative to the baseline diameter in the control condition:

“Note that the locomotion-evoked numbers reported in the text (unless otherwise stated), are relative to the control baseline, and thus include both the effects of the manipulation on the basal diameter and the evoked diameter. […] For comparison of the results obtained with these two approaches, during muscimol infusions, there was a small locomotion-evoked increase from the pre-locomotion baseline (15.37±8.11% vs. 5.14±11.09%, LME p<0.001, n=5 animals, 12 vessels) (Figure 6—figure supplement 2).”

We also note that using the change from the baseline will also give a misleading impression of the effects of several manipulations. For example, in the hsyn-Gq mice we see that the locomotion-evoked dilation is cut in half by CNO relative to the vehicle, (~18% with vehicle, ~9% with CNO, Figure 6—figure supplement 2E). A similar problem occurs with CaMKIIa-Gq mice with CNO infusion, which decreases the locomotion-evoked response (~19% vs. 9% , Figure 6—figure supplement 2G). This gives the reader the (erroneous) impression that *increasing* neural activity *decreases* the dilation, when actually it is reflecting the increased baseline diameter occurring during these manipulations. We think that it is clearer to add explanatory text to the muscimol result (which comes first in the paper), than to have to later explain how this measurement results in the potentially confusing results several times with hsyn-Gq and CaMKIIa-Gq DREADDs later on. To avoid burying the reader under numbers, we have split out the reported changes from baseline (both for pial and penetrating arterioles) in the “Comparison of pial and penetrating arteriole basal and evoked diameter changes” section and in Figure 6—figure supplement 2.